# TIMED DYNAMIC EXPANSION FOR CONTINUAL LEARNING

## ABSTRACT

Catastrophic forgetting remains a core challenge in continual learning, where parameter updates for new tasks interfere destructively with knowledge acquired from previous tasks. Dynamic expansion methods mitigate forgetting by inserting task-specific adapters, but rigid growth schedules often expand capacity unnecessarily on stable tasks while failing to protect against interference that arises later within a task. We propose Timed Dynamic Expansion (TIDE), a method that stabilises expansion by creating adapters only at the moments they are needed, preventing both wasted growth and destructive forgetting. This strategy improves training stability by limiting redundant modules, reduces memory overhead by avoiding expansion on tasks that do not conflict with prior knowledge, and ensures protection when forgetting arises unpredictably. At inference, TIDE combines adapter outputs through a Fisher Information–weighted gating mechanism to route information through the adapters most critical for retention. Experiments on standard CL benchmarks demonstrate that TIDE reduces forgetting, improves long-term retention, and achieves these gains with lower parameter growth than existing expansion methods. Code is available at https://anonymous.4open.science/r/TIDE-23B3/.

## 1 INTRODUCTION

Continual learning (Wang et al., 2024b; Jiang et al., 2025; Dohare et al., 2024) aims to enable models to acquire knowledge across a sequence of tasks without overwriting previously learned information. A central challenge in this setting is *catastrophic forgetting*, where parameter updates for new tasks interfere with representations important to earlier ones (Toneva et al., 2019; Yu et al., 2024). Dynamic architectural expansion (Yu et al., 2024; Wang et al., 2025) addresses this issue by introducing new parameters over time, allowing models to accommodate novel information while preserving past knowledge. Adapter-based methods (Rebuffi et al., 2017; Houlsby et al., 2019; Cui et al., 2025) have proven especially effective, as they insert lightweight modules into a frozen backbone, balancing flexibility with parameter efficiency.

Despite these advances, many existing expansion strategies follow rigid schedules or require observing a task in its entirety before allocating capacity (Yu et al., 2024; Zhou et al., 2025). Such approaches are inflexible: they may waste memory and dilute learning signals by over-allocating parameters to stable tasks, or fail to protect prior knowledge when interference arises unpredictably. Moreover, methods that rely on a complete pass over a task (Wang et al., 2025; Zhou et al., 2024) are impractical in streaming or task-free continual learning, where data arrives sequentially and revisiting prior tasks is infeasible. Consequently, both training stability and long-term retention are compromised, particularly when task difficulty and overlap fluctuate over time.

We propose *Timed Dynamic Expansion* (TIDE), a continual learning framework that expands capacity when forgetting emerges. TIDE monitors interference continuously during training and introduces new adapters only when statistically significant forgetting is detected. This ensures that tasks prone to interference may receive multiple adapters over time, while stable tasks may require none at all. At inference, TIDE activates all adapters associated with a task and combines their outputs using a Fisher-informed gating mechanism, which prioritises modules most critical for retention. By aligning expansion and routing with the actual dynamics of forgetting, TIDE stabilises training, avoids unnecessary growth, and improves long-term performance.

Our contributions are threefold. First, we introduce a mechanism for dynamic adapter allocation based on detecting forgetting events, ensuring that new capacity is added only when negative interference is significant. Second, we propose a Fisher-informed inference scheme that aggregates multiple adapters per task, allowing the model to recover and stabilise performance throughout training. Third, we demonstrate that continual monitoring eliminates the need for rigid growth schedules, enabling asymmetric and time-sensitive expansion that achieves both stability and parameter efficiency across tasks.

## 2 PRELIMINARIES

Here, we define the learning settings and notations used to describe our techniques.

**Continual Learning.** Let $\mathcal{T} = \{\mathcal{T}_1, \mathcal{T}_2, \ldots, \mathcal{T}_K\}$ denote a sequence of $K$ tasks in a continual learning setting. Each task $\mathcal{T}_k$ is defined by a dataset $\mathcal{D}_k = \{(\mathbf{x}_i, y_i)\}_{i=1}^{n_k}$, where $\mathbf{x}_i \in \mathcal{X}$ are input samples and $y_i \in \mathcal{Y}_k$ are labels drawn from the task-specific label space $\mathcal{Y}_k$.

We address a Class-Incremental Learning (CIL) setting. CIL assumes no task identity is available at test time. Each task introduces a disjoint subset of classes $\mathcal{Y}_j \cap \mathcal{Y}_k = \emptyset$ for all $j \neq k$, and the model must learn a function $f_\theta : \mathcal{X} \to \bigcup_{k=1}^{K} \mathcal{Y}_k$ that can classify among all previously observed classes. Without access to task identity, the model must integrate all class knowledge into a single unified output space and resolve class boundaries across tasks. This makes CIL a more difficult setting, as it requires the model to retain discriminative features for all classes without relying on external contextual information to reduce the output domain. The core challenge is to prevent catastrophic forgetting, preserving performance on past tasks while integrating knowledge from new ones without direct access to prior data.

**Fisher Information.** The Fisher Information measures the sensitivity of the model likelihood $p(y|\mathbf{x}; \boldsymbol{\theta})$ to changes in parameters. For a parameter vector $\boldsymbol{\theta}$, the Fisher Information Matrix (FIM) is defined as the expected outer product of log-likelihood gradients:

$$\mathcal{F} = \mathbb{E}_{(\mathbf{x},y)\sim p} \left[ \nabla_{\boldsymbol{\theta}} \log p(y|\mathbf{x}; \boldsymbol{\theta}) \, \nabla_{\boldsymbol{\theta}} \log p(y|\mathbf{x}; \boldsymbol{\theta})^\top \right].$$

It provides a measure of parameter importance with respect to the predictive distribution (Kirkpatrick et al., 2017). Since calculating the full $\mathcal{F}$ is impractical, we approximate importance using its trace, which correlates with the behaviour of the full matrix while remaining computationally efficient (Achille et al., 2018):

$$\text{tr}(\mathcal{F}) = \mathbb{E}_{\mathbf{x}\sim p} \left[ \mathbb{E}_{\hat{y}\sim p(\hat{y}|\mathbf{x};\boldsymbol{\theta})} \|\nabla_{\boldsymbol{\theta}} \log p(\hat{y}|\mathbf{x}; \boldsymbol{\theta})\|^2 \right].$$

Details of the batch-wise Fisher trace approximation are given in Appendix D.

## 3 RELATED WORK

Continual learning seeks to enable models to learn new tasks without catastrophically forgetting previously acquired knowledge (Wang et al., 2024a; Dohare et al., 2024). Existing approaches can be broadly categorised into *regularisation-based methods*, which constrain parameter updates using importance measures (Kirkpatrick et al., 2017; Li & Hoiem, 2017), *replay-based methods*, which store or generate past samples to stabilise training (Rebuffi et al., 2017; Chaudhry et al., 2019b), and *dynamic architectural methods*, which introduce new capacity to preserve old knowledge while adapting to new tasks (Rusu et al., 2016; Yoon et al., 2017). Our work falls into the third category, building on recent adapter-based expansions.

**Adapter-Based Expansion.** Adapters (Rebuffi et al., 2017; Houlsby et al., 2019) are lightweight task-specific modules for transfer learning, and have since been widely adopted for continual learning due to their parameter efficiency (Cui et al., 2025). By freezing the backbone and only updating adapter parameters, models can allocate capacity selectively while limiting interference across tasks.

Recent works extend this idea by expanding adapter sets over time. Mixture-of-experts formulations, for example, introduce new adapters to improve flexibility and isolate task-specific knowledge (Zhou et al., 2024; Yu et al., 2024). To control growth, these methods decide whether and how much to

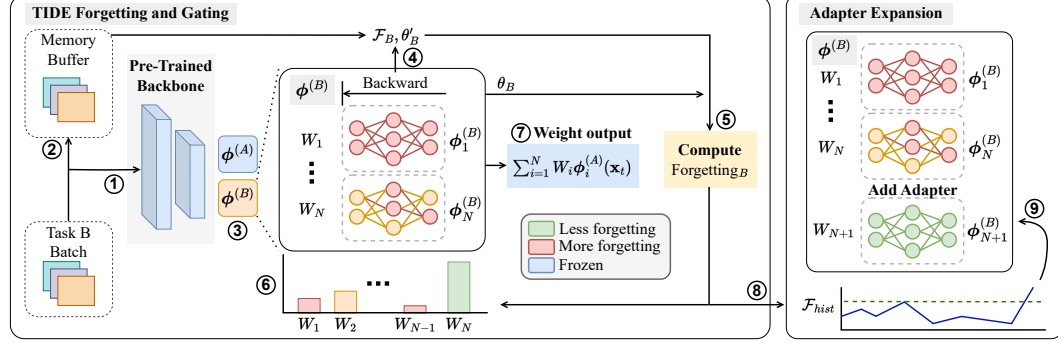

Figure 1: **Overview of the TIDE architecture** during training on Task B. (1) New batch is input to pre-trained layers. (2) A sample of incoming data is stored in the memory buffer. (3) The dynamic adapter layer, where adapters trained on a previous Task A are frozen and Task B adapters are trained. Currently, Task B has $N$ assigned adapters, each with corresponding weights $W$. Current weights $\theta_B$ are stored, and (4) updated weights $\theta'_B$ and Fisher trace $\mathcal{F}_B$ are acquired during the backward pass. (5) Forgetting$_B$ is calculated (Eq. 1). (6) Weights for each adapter are scaled with the Fisher trace and used to (7) compute adapter outputs. (8) Fisher trace is tracked in $\mathcal{F}_{\text{hist}}$, and if the new score is statistically out-of-distribution, (9) a new adapter is added for Task B.

expand at each task boundary, often guided by heuristics (Lee et al., 2024) or task-level distributional metrics (Wang et al., 2025). This ensures that parameters do not grow unbounded, but it also constrains expansion to a single decision per task. Such boundary-informed approaches implicitly assume that the interference introduced by a task is both uniform and stable from the point of its introduction onward. In practice, however, forgetting can emerge earlier within a task, later than the transition, or fluctuate multiple times during its progression. As a result, boundary-based strategies may under-allocate capacity for difficult tasks that interfere immediately, or over-allocate for tasks that only partially diverge from prior knowledge. This motivates the need for approaches that decouple expansion from task boundaries and instead trigger growth dynamically in response to observed forgetting.

**Forgetting.** Catastrophic forgetting refers to the tendency of neural networks to overwrite representations learned on previous tasks when trained sequentially. This has been studied through the lens of synaptic plasticity (Kirkpatrick et al., 2017), functional drift (Mirzadeh et al., 2020), and representational stability (Ramasesh et al., 2021). Several techniques have been proposed to mitigate forgetting, including regularisation-based methods (Zenke et al., 2017; Kirkpatrick et al., 2017), replay-based methods (Rebuffi et al., 2017; Chaudhry et al., 2019b), and architectural isolation (Yoon et al., 2017). While most approaches treat forgetting as an aggregate effect, recent work has begun to quantify forgetting at the level of individual tasks or examples (Toneva et al., 2019; Chaudhry et al., 2019a). Our method builds on this direction by defining a Fisher-weighted forgetting score, enabling fine-grained measurement of interference and triggering expansion only in response to statistically significant forgetting events. This allows the model to prioritise capacity allocation based on empirical evidence rather than heuristic schedules.

## 4  TIMED DYNAMIC EXPANSION

Timed Dynamic Expansion (TIDE) is a continual learning technique designed to improve the effectiveness of parameter expansion by targeting both inter-task and intra-task interference. In continual learning, tasks can interfere with one another (inter-task) or with earlier phases of themselves over time (intra-task), leading to degraded performance. TIDE addresses this by introducing newly initialised parameters for a task only when that task is actively experiencing forgetting. First, we define a forgetting measure that quantifies the model's deviation from a task-specific solution, weighted by the importance of each parameter as estimated by the FIM. Second, we propose TIDE, a mechanism that selectively expands model capacity by appending new, trainable parameters for a task only when its forgetting measure indicates significant degradation. Finally, we analyse the computational

complexity of our approach, showing that TIDE enables targeted parameter growth with bounded computational and memory overhead.

## 4.1 FORGETTING DETECTION

Let a model $\mathcal{M}$ have parameters $\boldsymbol{\theta} = (\theta_1, \theta_2, \ldots, \theta_n)$ and a corresponding task loss function $\mathcal{L}(\boldsymbol{\theta}, \mathcal{D}_A)$ for task $A$, where $\mathcal{D}_A$ is the data distribution for task $A$. The objective of learning a task is to minimise the loss function, i.e., find $\boldsymbol{\theta}^*(A)$ such that:

$$\boldsymbol{\theta}^*(A) = \arg\min_{\boldsymbol{\theta}} \mathcal{L}(\boldsymbol{\theta}, \mathcal{D}_A).$$

Let the model be trained on task $A$ and then exposed to a new task $t$. After learning task $t$, the parameters $\boldsymbol{\theta}$ are updated to $\boldsymbol{\theta}^{(t)}$, where $\boldsymbol{\theta}^{(t)}$ is the parameter vector after task $t$ has been learned. We define the change in the parameter $\theta_i$ due to learning the new task $t$ as:

$$\Delta\theta_i = \theta_i^{*(A)} - \theta_i^{(t)}.$$

Let $\mathcal{F}_{ij}^{(A)}$ represent the FIM for task $A$ corresponding to the $i$-th and $j$-th parameters. The FIM captures the curvature of the loss function around the optimal parameters $\boldsymbol{\theta}^*(A)$. Specifically, the FIM can be approximated as the second derivative of the loss function with respect to the parameters:

$$\mathcal{F}_{ij}^{(A)} = -\mathbb{E}_{(\mathbf{x},y)\sim p} \left[ \frac{\partial^2 \log p(y|\mathbf{x};\boldsymbol{\theta})}{\partial\theta_i\partial\theta_j} \right] \Bigg|_{\theta=\theta^*(A)}.$$

The sensitivity of the loss function to changes in the parameters is related to the FIM. The first-order approximation of the change in the loss can be expressed as:

$$\Delta\mathcal{L} = \sum_{i=1}^n \frac{\partial\mathcal{L}(\boldsymbol{\theta})}{\partial\theta_i}\Delta\theta_i + \frac{1}{2}\sum_{i,j=1}^n \Delta\theta_i\Delta\theta_j \frac{\partial^2\mathcal{L}(\boldsymbol{\theta})}{\partial\theta_i\partial\theta_j}.$$

For small perturbations in the parameters $\|\Delta\theta_i\| \ll 1$, the leading-order change in loss is driven by the second derivative, which is related to the FIM. Specifically:

$$\Delta\mathcal{L} \approx \frac{1}{2}\sum_{i,j=1}^n \mathcal{F}_{ij}^{(A)}\Delta\theta_i\Delta\theta_j.$$

For parameters $i$ with high Fisher information $\mathcal{F}_{ij}^{(A)}$, the loss is highly sensitive to changes in $\theta_i$. Therefore, large shifts $\Delta\theta_i$ cause a significant increase in $\Delta\mathcal{L}$, with the effect amplified for parameters with higher FIM values. In continual learning, such changes can increase the task loss for $A$, contributing to forgetting. The forgetting score $F_A(k)$ at time step $k$, defined as:

$$F_A(k) = \sum_i \mathcal{F}_i^{(A)}(\theta_i^{*(A)} - \theta_i^{(t)})^2, \tag{1}$$

where $\mathcal{F}_i^{(A)}$ denotes the trace approximation of the Fisher score, captures this phenomenon by quantifying the weighted sum of parameter shifts, with larger shifts in more important parameters (those with high $\mathcal{F}$) resulting in a higher forgetting score. Here, we use the trace approximation of the full $\mathcal{F}$ for computational efficiency.

Assume introduce $\mathcal{N}_\theta$ new parameters to model $\mathcal{M}$, yielding a new parameter vector:

$$\boldsymbol{\theta}' = (\boldsymbol{\theta}, \boldsymbol{\phi}) \in \mathbb{R}^{n+\mathcal{N}_\theta},$$

where $\boldsymbol{\phi} \in \mathbb{R}^{\mathcal{N}_\theta}$ are newly added parameters, initialized such that they do not affect performance on task $A$; formally, for all $(\mathbf{x}, y) \sim \mathcal{D}_A$, we assume:

$$\frac{\partial\mathcal{L}(\boldsymbol{\theta}, \mathcal{D}_A)}{\partial\phi_j} = 0, \quad \forall j \in \{1, \ldots, \mathcal{N}_\theta\},$$

and similarly,

$$\frac{\partial^2\mathcal{L}(\boldsymbol{\theta}, \mathcal{D}_A)}{\partial\theta_i\partial\phi_j} = 0, \quad \forall i, j.$$

This implies the FIM $\mathcal{F}^{(A)}$ is block-diagonal with respect to the extended parameter space:

$$\mathcal{F}^{(A)} = \begin{bmatrix} \mathcal{F}^{(A)}_{\boldsymbol{\theta}} & 0 \\ 0 & 0 \end{bmatrix}.$$

Thus, for updates confined to $\phi$, we have

$$F'_A(k+1) = \sum_i \mathcal{F}^{(A)}_i \big(\theta^{*(A)}_i - \theta'^{(t)}_i\big)^2 \;\le\; F_A(k), \tag{2}$$

where $F'_A(k+1)$ denotes the forgetting score after parameter expansion into the augmented space $(\boldsymbol{\theta}, \boldsymbol{\phi})$. This suggests that parameter expansion during forgetting is more effective. Lemma 1 provides the formal second-order justification for Equation 2 using the full FIM, and we find that the expanded parameter space contains directions that improve the new task with bounded forgetting on previous tasks by Lemma 2.

## 4.2 ADAPTER EXPANSION

The key principle of TIDE is that expansion should occur *when* forgetting arises instead of at task boundaries. Given the forgetting score in Equation 1, TIDE continuously monitors past tasks during training on the current task and allocates new adapters when interference is statistically significant. This allows expansion to follow the dynamics of forgetting rather than externally defined tasks.

Let $\mathcal{T}_{\text{past}}$ denote the set of previously seen tasks. At each training step $k$, TIDE estimates $F_A(k)$ for every $A \in \mathcal{T}_{\text{past}}$ using a small batch of data from $\mathcal{D}_A$. If the forgetting score for any task $A$ exceeds the expected distribution under a significance level $\alpha$, the model appends a task-specific adapter $\phi^{(A)}$. These adapters are lightweight bottleneck modules (Houlsby et al., 2019), defined as

$$\phi^{(A)}(h) = h + f(h\theta_{\text{down}})\theta_{\text{up}},$$

with parameters $\theta_A = \{\theta^A_{\text{down}}, \theta^A_{\text{up}}\}$ for down and up projection respectively, and nonlinearity $f(\cdot)$. In our implementation for image classification, adapters are inserted after the attention and feed-forward sub-layers of a vision transformer backbone. One adapter is instantiated prior to training, and all subsequent adapters are created by the TIDE process. For Multimodal Large Language Model (MLLM) implementations, adapters are inserted at Multi-Layer Perceptron (MLP) layers.

Expansion is triggered when the batch-wise forgetting score of some task $A$ is statistically out-of-distribution relative to historical scores $\mathcal{F}_{\text{hist}}$:

$$\exists A \in \mathcal{T}_{\text{past}} \quad \text{s.t.} \quad p\big(F_A(k) \mid \mathcal{F}_{\text{hist}}\big) < \alpha \quad \Rightarrow \quad \text{Add adapter } \phi^{(A)}_{N+1}.$$

Here, $\alpha$ is a confidence level controlling responsiveness. We provide details of the smoothing of $\mathcal{F}_{\text{hist}}$ in Appendix D. Figure 1 illustrates this process as adapters are created in response to forgetting events. Smaller values of $\alpha$ make TIDE less prone to growth, while larger values are more sensitive to expansion. Theorem 1 shows that the risk associated with expansion decisions is a strictly convex function of $\alpha$, ensuring the existence of a single, well-defined optimal threshold.

**Theorem 1.** *Let $\theta \in \mathbb{R}^d$ be a fixed parameter. Two estimators $\hat{\theta}_1, \hat{\theta}_2$ satisfy $\mathbb{E}\|\hat{\theta}_i\|^2_2 < \infty$. Define errors $e_i = \hat{\theta}_i - \theta$, biases $b_i = \mathbb{E}[e_i]$, and covariance blocks $\Omega_{ij} = \text{Cov}(e_i, e_j)$ for $i, j \in \{1, 2\}$. Let $W \in \mathbb{R}^{d \times d}$ be symmetric positive definite and measure risk by $R(\alpha) = \mathbb{E}\big[\|\hat{\theta}(\alpha) - \theta\|^2_W\big]$ where $\hat{\theta}(\alpha) = \alpha\hat{\theta}_1 + (1-\alpha)\hat{\theta}_2$ and $\|x\|^2_W = x^\top W x$. Assume*

$$D := \text{tr}\big(W(\Omega_{11} + \Omega_{22} - \Omega_{12} - \Omega_{21})\big) + (b_1 - b_2)^\top W(b_1 - b_2) > 0.$$

*The function $R(\alpha)$ is a strictly convex quadratic in $\alpha$ with unique unconstrained minimizer*

$$\alpha^\star = \frac{\text{tr}\big(W(\Omega_{22} - \Omega_{12})\big) + b_2^\top W(b_2 - b_1)}{D}.$$

*If $\alpha$ is constrained to $[0, 1]$, the optimal setting is the projection*

$$\alpha_{\text{opt}} = \Pi_{[0,1]}(\alpha^\star) = \min\{1, \max\{0, \alpha^\star\}\}.$$

*Proof.* The complete proof is provided in Appendix G, and an empirical validation to acquire $\alpha = 0.4$ for our experiments is shown in Appendix H.

This result justifies using $\alpha$ as a means to control expansion, providing stability by bounding false positives while ensuring responsiveness to forgetting. This aligns with the goal of avoiding both over-allocation and under-allocation by adapting growth to observed interference.

**Bounding Long-Term Growth.** While adaptive expansion improves plasticity, unchecked growth risks undermining stability. To avoid this, TIDE sets an upper bound on the number of adapters per task. This limit balances the diminishing returns of adding capacity against the quadratic stability cost of maintaining too many modules. Theorem 2 formalises this stability–plasticity trade-off by showing that the long-term risk is a strictly convex function of the adapter count, yielding a unique optimal cap on the number of adapters per task. This provides a principled way to prevent runaway growth.

**Theorem 2.** *Assume tasks $t = 1, \ldots, T$ and constants $a_t > 0$, $\gamma > 0$, $s > 0$, where $a_t$ is the plasticity scale, $\gamma$ is the diminishing-returns rate, and $s$ is the stability scale, exist such that for any adapters added $m \geq 0$, $\mathcal{R}_t(m) \leq a_t e^{-\gamma m} + s m^2$. Let $A = \frac{1}{T} \sum_{t=1}^{T} a_t$ and define $R(M) = A e^{-\gamma M} + s M^2$. Then $R(M)$ is strictly convex on $[0, \infty)$ with unique minimizer*

$$M^\star = \frac{1}{\gamma} W\left(\frac{\gamma^2 A}{2s}\right),$$

*where $W$ is the Lambert $W$-function. For integer caps $M \in \{0, \ldots, M_{\max}\}$, the optimal cap is*

$$M_{\text{opt}} = \arg \min_{m \in \{0, \ldots, M_{\max}\}} \{A e^{-\gamma m} + s m^2\},$$

*or, equivalently, the projection of $M^\star$ onto $\{0, \ldots, M_{\max}\}$.*

*Proof.* The complete proof is provided in Appendix I, and an empirical validation to acquire $M = 10$ for our experiments is shown in Appendix J.

**Forgetting Estimation.** Finally, to estimate forgetting without access to full datasets, TIDE maintains a small memory buffer $\mathcal{B}_A$ with a maximum of $N$ samples for each task $A \in \mathcal{T}_{\text{past}}$. We use reservoir sampling to ensure the buffer contains samples that reflect the distribution of the data stream. A mini-batch from $\mathcal{B}_A$ is used both to compute the forgetting score and to reuse the FIM estimated at task completion. The mini-batches are not used to train the model, avoiding the extra complexity of full rehearsal methods. This avoids recomputing the FIM from scratch at each training step. Pseudocode for the full TIDE adapter expansion procedure is given in Appendix A, and the theoretical complexity of the method is discussed in Appendix C.

## 4.3 ADAPTER GATING MECHANISM

To accommodate repeated forgetting within a task, TIDE permits multiple adapters per task. Let $\{\phi_i^{(A)}\}_{i=1}^{N}$ denote the set of $N$ adapters assigned to task $A$. During inference on task $A$, adapter outputs are combined using a Fisher-informed gating mechanism. Specifically, each adapter $\phi_i^{(A)}$ is assigned a weight $W_i$ based on its historical importance, measured by the Fisher Information corresponding to its parameters at the time of insertion:

$$W_i = \frac{\mathcal{F}_i}{\sum_{j=1}^{N} \mathcal{F}_j}$$

where $\mathcal{F}_i$, is the Fisher trace of $\phi_i^{(A)}$. Given an input $\mathbf{x}_t$ for task $A$, the final adapter-weighted representation for prediction is computed as a weighted sum of the outputs of all $N$ adapters:

$$\mathbf{y}_t = \sum_{i=1}^{N} W_i \phi_i^{(A)}(\mathbf{x}_t).$$

The weights $\{W_i\}$ are normalised to sum to 1 and reflect the relative preservation priority of each adapter. Higher Fisher traces correspond to larger weights, biasing the prediction toward more critical parameter regions. This gating strategy allows the model to maintain multiple specialised recovery modules for a given task and dynamically modulate their influence based on their estimated contribution to task retention. Appendix B gives a pseudocode implementation of the adapter gating process.

Table 1: Classification accuracy (%) of various continual learning baselines on benchmark image datasets. Blue values indicate the highest accuracy for the dataset, red values indicate the second-highest accuracy.

| Method | CIFAR-100 | | CUB | | IN-R | | IN-A | | IN-1k | | CORe50 | |
|---|---|---|---|---|---|---|---|---|---|---|---|---|
| | $\bar{\mathcal{A}}\uparrow$ | $\mathcal{A}_B\uparrow$ | $\bar{\mathcal{A}}\uparrow$ | $\mathcal{A}_B\uparrow$ | $\bar{\mathcal{A}}\uparrow$ | $\mathcal{A}_B\uparrow$ | $\bar{\mathcal{A}}\uparrow$ | $\mathcal{A}_B\uparrow$ | $\bar{\mathcal{A}}\uparrow$ | $\mathcal{A}_B\uparrow$ | $\bar{\mathcal{A}}\uparrow$ | $\mathcal{A}_B\uparrow$ |
| LwF | 46.36 | 41.21 | 49.45 | 32.36 | 39.35 | 26.11 | 37.57 | 26.94 | 33.37 | 27.51 | 47.46 | 33.44 |
| SDC | 68.21 | 63.05 | 70.42 | 57.27 | 52.99 | 42.29 | 47.91 | 26.63 | 62.48 | 50.47 | 69.55 | 55.92 |
| L2P | 85.94 | 77.93 | 67.05 | 66.25 | 56.53 | 52.92 | 49.39 | 41.71 | 63.72 | 52.50 | 73.36 | 64.69 |
| DP | 87.87 | 81.15 | 71.77 | 70.34 | 62.57 | 58.74 | 55.55 | 47.29 | 68.92 | 57.89 | 75.33 | 65.92 |
| CODA-P | 89.11 | 81.96 | 84.00 | 73.37 | 64.42 | 55.08 | 47.22 | 35.02 | 70.63 | 58.79 | 73.93 | 68.08 |
| SimpleCIL | 87.57 | 81.92 | 82.82 | 72.61 | 62.27 | 54.07 | 60.01 | 49.26 | 68.35 | 57.89 | 75.57 | 71.53 |
| InfLoRA | 87.67 | 81.27 | 91.22 | 82.39 | 59.11 | 52.42 | 59.75 | 46.20 | 84.22 | 83.08 | 75.43 | 72.03 |
| SEDEM | 87.58 | 81.24 | 91.59 | 83.05 | 60.01 | 53.42 | 61.41 | 54.28 | 84.11 | 83.10 | 83.04 | 77.93 |
| MoE | 88.46 | 82.34 | 91.58 | 82.88 | 61.08 | 54.74 | 62.76 | 55.47 | 82.71 | 83.05 | 82.32 | 76.37 |
| ADAM | 90.65 | 85.15 | 92.21 | 86.34 | 64.33 | 60.47 | 63.47 | 49.37 | 83.98 | 83.11 | 84.75 | 77.37 |
| EASE | **91.51** | 85.80 | 92.23 | 86.81 | 78.31 | 70.58 | 65.34 | 55.04 | 84.29 | 83.13 | 87.24 | 80.85 |
| SEMA | 91.29 | **86.01** | 92.35 | 87.51 | 78.34 | 71.19 | 64.20 | 53.45 | 84.25 | 83.04 | 88.58 | 82.41 |
| TIDE | 90.88 | 85.41 | **92.51** | **87.56** | **80.40** | **72.61** | **69.24** | **61.00** | **84.37** | **83.16** | **90.99** | **84.58** |

# 5 EXPERIMENTS

We compare TIDE with SOTA continual learning algorithms on six benchmark image datasets and two MLLM datasets to assess incremental and overall learning ability. Additionally, we provide an ablation study to investigate the contribution of key components of TIDE. We also analyse the sensitivity of our method to its hyperparameters. Finally, we compare the parameter growth of expansion strategies.

**Datasets.** We follow Zhou et al. (2024); Tong et al. (2025) to evaluate the performance of each method on CIFAR-100 (Krizhevsky et al., 2009), CUB (Dataset, 2011), ImageNet-R (IN-R) (Hendrycks et al., 2021a) and ImageNet-A (IN-A) (Hendrycks et al., 2021b). We also use the continual learning benchmark CORe50 (Lomonaco & Maltoni, 2017). These datasets contain typical CIL benchmarks and out-of-distribution datasets that have a large domain gap with ImageNet, which is used for training on the ViT-B/16-IN21K backbone. For robust evaluation, we also provide results over ImageNet-1k (IN-1k) (Russakovsky et al., 2015). Beyond image classification, we demonstrate an application for TIDE with MLLMs on benchmark datasets VQAv2 (Goyal et al., 2017) and SciQA-IMG (Lu et al., 2022). A thorough description of our benchmark datasets with preprocessing details is given in Appendix K.

**Baselines.** We choose SOTA CIL continual learning techniques with pre-trained backbones for comparison: LwF (Li & Hoiem, 2017), SDC (Yu et al., 2020), L2P (Wang et al., 2022b), Dual-Prompt (DP) (Wang et al., 2022a), CODA-P (Smith et al., 2023), SimpleCIL (Zhou et al., 2025), InfLoRA (Liang & Li, 2024), SEDEM (Ye & Bors, 2023), MoE (Yu et al., 2024), ADAM (Zhou et al., 2025), EASE (Zhou et al., 2024) and SEMA (Wang et al., 2025). All baselines are evaluated under the same class-incremental setting, with consistent task boundaries and training splits. Unless otherwise specified, we reimplement methods based on published configurations. Image classification methods are implemented with the same pre-trained backbone, ViT-B/16-IN21K.

**Reproducibility.** All experiments were implemented in PyTorch with CUDA compatibility and executed on NVIDIA H200 GPUs. We ensure consistency in architectural design, data preprocessing and training hyperparameters ($\alpha = 0.4, N = 100, M = 10$) across all baselines and ablation studies. To control for the sensitivity of continual learning models to task ordering and initialisation (Krueger & Dayan, 2009), each dataset is ordered into class-incremental tasks, with the class sequence randomly shuffled in each trial. For each of the five random class orderings, we repeat the training process five times with different random seeds. This ensures robustness to dataset ordering and stochastic variation in training.

**Classification Accuracy.** Table 1 presents the classification accuracy of various continual learning methods with pre-trained backbone ViT-B/16-IN21K across six benchmark image datasets. We use

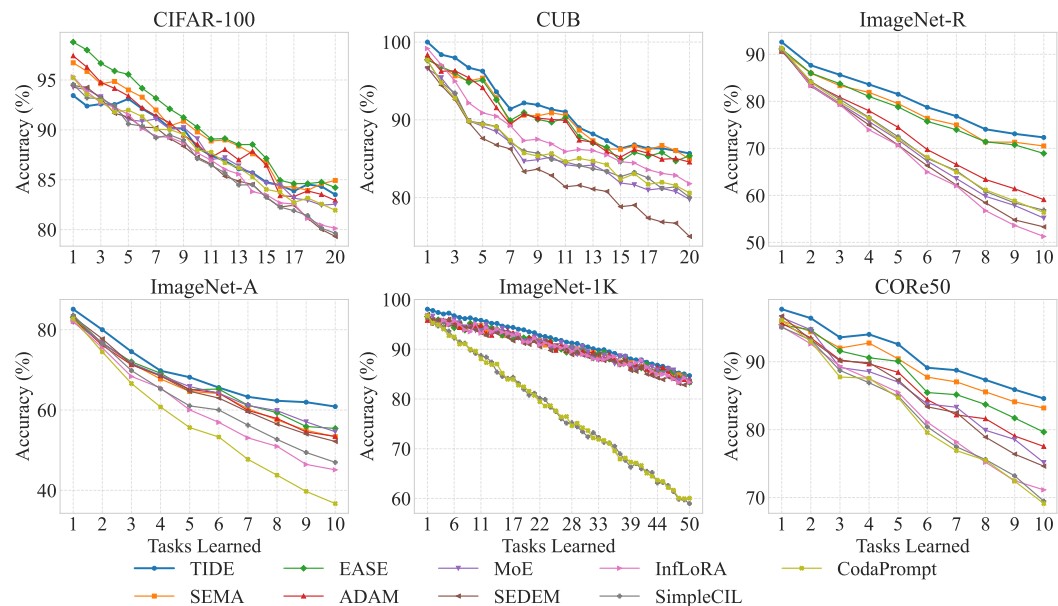

Figure 2: Incremental accuracy of baseline methods on continual learning benchmarks.

two metrics to validate our results; $\bar{\mathcal{A}}$ represents the average incremental accuracy during training, and $\mathcal{A}_B$ represents the average accuracy over all tasks after training. On five out of six datasets, TIDE achieves the highest accuracy, outperforming the next best baseline by up to 5.96% (on IN-A). IN-R and IN-A contain frequently misclassified examples, suggesting that TIDE's expansion technique is particularly effective for more difficult tasks. TIDE also yields SOTA performance on CUB and CORe50, which contain easier tasks, demonstrating the robustness and generality of TIDE, particularly in its ability to scale across both simple and complex continual learning benchmarks.

Figure 2 reports the incremental performance trend of SOTA methods during training. We find that the rate of accuracy decline as more tasks are learned is lower for TIDE, particularly on IN-R and IN-A, suggesting that our approach for adapter expansion helps to alleviate forgetting throughout training. The task-specific performance of adapters, and empirical examples of when they are added by TIDE, are provided in Appendix M.

**Ablation Study.** We evaluate the impact of various adapter gating strategies on the performance of TIDE in Table 2. The TIDE gating strategy achieves the highest performance. Linear interpolation, which averages adapter outputs based on fixed weights, shows the weakest performance, suggesting it lacks the flexibility to account for task-specific dynamics. Attention-based gating, which computes soft weights conditioned on inputs, performs moderately better but still trails behind TIDE. Binary masking, which applies hard selection among adapters, underperforms due to its limited expressivity and coarse decision-making. Figure 3 also provides results for the ablation of the adapter component. At $\alpha = 0$,

Table 2: TIDE performance ($\mathcal{A}_B$) under different adapter gating strategies: Linear Interpolation, Attention-Based and Binary Masking.

| Gating | CIFAR-100 | IN-A |
|---|---|---|
| Linear | 81.13 | 51.73 |
| Attention | 80.28 | 53.21 |
| Binary | 78.92 | 52.10 |
| TIDE | **85.41** | **55.26** |

adapters cannot be created because the probability of a forgetting score being observed can never be less than $0$. In this case, TIDE performs identically to a static model, as there are no trainable parameters added during training. Table 3 shows the ablation of TIDE's adapter expansion strategy, and compares performance with three naive and heuristic-based sampling methods. Task Expansion, Random Expansion, and Even Expansion, which are detailed in Appendix O, achieve similar $\mathcal{A}_B$ at constant parameter growth. Even Expansion is marginally more robust than Random Expansion and often ties or slightly improves over Task Expansion, suggesting that distributing plasticity across a task helps to capture late interference without harming early adaptation. TIDE matches or exceeds the best fixed-timing baselines while using less capacity. Results for the computational complexity, runtime and memory requirements of TIDE against other baselines are provided in Appendix L.

Table 3: Comparison of adapter expansion strategies. Params denotes the number of trainable parameters per learning setting in millions. Adapter capacity $M = 10$ for all expansion strategies.

| Dataset | Task Expansion | | Random Expansion | | Even Expansion | | TIDE | |
|---|---|---|---|---|---|---|---|---|
| | Params | $\mathcal{A}_B \uparrow$ | Params | $\mathcal{A}_B \uparrow$ | Params | $\mathcal{A}_B \uparrow$ | Params | $\mathcal{A}_B \uparrow$ |
| CIFAR-100 | 1.066 | 85.26 | 1.066 | 85.11 | 1.066 | 85.13 | 0.512 | **85.41** |
| CUB | 1.066 | 87.18 | 1.066 | 87.08 | 1.066 | 87.03 | 0.625 | **87.56** |
| ImageNet-R | 1.904 | 72.28 | 1.904 | 72.23 | 1.904 | 72.25 | 0.852 | **72.61** |
| ImageNet-A | 1.904 | 60.93 | 1.904 | 60.41 | 1.904 | 60.38 | 1.250 | **61.00** |
| ImageNet-1k | 1.904 | 83.00 | 1.904 | 83.01 | 1.904 | 83.05 | 1.189 | **83.16** |
| CORe50 | 1.066 | 84.02 | 1.066 | 83.98 | 1.066 | 83.91 | 1.050 | **84.58** |

**Sensitivity.** Figure 3 shows the sensitivity of TIDE to the forgetting threshold $\alpha$ used to decide when to instantiate new adapters. A high $\alpha$ causes frequent expansions, resulting in too many adapters per task. This overallocation not only deteriorates performance due to parameter saturation, but also complicates the gating mechanism, since routing becomes less reliable when many adapters are available for the same task. Conversely, if $\alpha$ is set too low, expansions are suppressed even when forgetting is substantial, which risks underfitting new tasks and leaving past knowledge unprotected. Further sensitivity studies on buffer size $N$ and adapter cap $M$ are provided in Appendix P.

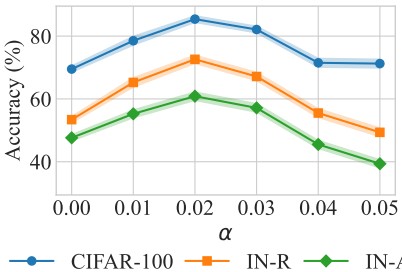

Figure 3: Sensitivity of TIDE to forgetting threshold $\alpha$.

Table 4: TIDE and naive adapter expansion for training on MLLM benchmarks with LLaVA-1.5-7B.

| Method | VQAv2 | | | SciQA-IMG | | |
|---|---|---|---|---|---|---|
| | Params | $\bar{\mathcal{A}} \uparrow$ | $\mathcal{A}_B \uparrow$ | Params | $\bar{\mathcal{A}} \uparrow$ | $\mathcal{A}_B \uparrow$ |
| LLaVA-1.5-7B | - | - | 80.00 | - | - | 71.60 |
| + Task Expansion | 23.99 | 81.57 | 81.91 | 10.04 | 72.01 | 72.17 |
| + TIDE | 18.58 | **81.54** | **82.01** | 9.11 | **72.50** | **72.81** |

**Application.** We demonstrate the flexibility of our method, applying TIDE to LLaVA-1.5-7B (Liu et al., 2023), keeping all pre-trained layers frozen. We use the VQAv2 MLLM dataset for training the adapter layer, and provide further details on the dataset and its implementation for continual learning in Appendix K. As shown in Table 4, TIDE limits parameter growth by approximately 22.6% while maintaining performance, demonstrating that TIDE can be effectively integrated for various applications.

## 6 CONCLUSION

We addressed the challenge of dynamic expansion in continual learning, where fixed or boundary-driven schedules often fail to balance stability and plasticity across tasks. We proposed TIDE, a method that selectively introduces adapters only when tasks exhibit statistically significant forgetting. Unlike heuristic or boundary-based expansion strategies, TIDE grows model capacity only when needed, which both prevents wasted parameters and improves retention. At inference, TIDE incorporates a Fisher-informed gating mechanism to combine multiple adapters per task, ensuring that predictions prioritise historically important parameters. Across benchmark datasets, TIDE consistently outperformed or matched SOTA baselines, achieving up to 5.96% higher accuracy on challenging benchmarks while using substantially fewer trainable parameters. These results highlight the advantages of decoupling expansion from task boundaries. By linking architectural growth directly to observed forgetting, TIDE demonstrates how continual learners can regulate their own capacity in response to forgetting.

## REPRODUCIBILITY STATEMENT

We ensure reproducibility by providing full experimental details, along with complete proofs in the appendices. All datasets used in our experiments are referenced and publicly available. Code implementation of our experiments is available at `https://anonymous.4open.science/r/TIDE-23B3/`.

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

APPENDIX

## A   ALGORITHM FOR ADAPTER EXPANSION

This section provides a pseudocode implementation for the adapter expansion procedure in TIDE. Algorithm 1 shows how TIDE monitors forgetting scores across past tasks and triggers adapter creation when statistically significant interference is detected. At each training step, forgetting scores are estimated using stored Fisher diagonals and memory buffers, compared to historical distributions, and used to decide whether to expand. This algorithm highlights that expansion is task- and time-sensitive, and that adapters are introduced only when forgetting is observed.

---

**Algorithm 1** TIDE Adapter Expansion

---

**Require:** Current task $t$; past tasks $\mathcal{T}_{\text{past}}$; memory buffers $\{\mathcal{B}_A\}$; reference parameters $\{\theta_A^*\}$; Fisher diagonals $\{\mathcal{F}_A\}$; forgetting history $\mathcal{F}_{\text{hist}}$; significance threshold $\alpha$
1: **for** each training step $k$ **do**
2:     **for** each $A \in \mathcal{T}_{\text{past}}$ **do**
3:         Sample batch $(x_i, y_i) \sim \mathcal{B}_A$
4:         Compute current parameters $\theta_k$
5:         $F_A(k) \leftarrow \sum_j \mathcal{F}_A[j] \cdot (\theta_k[j] - \theta_A^*[j])^2$
6:         Estimate p-value: $p_A \leftarrow p(F_A(k), \mathcal{F}_{\text{hist}})$
7:         **if** $p_A < \alpha$ **then**
8:             Add new adapter $\phi_{N+1}^{(A)}$:
9:                 $\theta_{\text{down}}^{(A)} \in \mathbb{R}^{d \times r}, \theta_{\text{up}}^{(A)} \in \mathbb{R}^{r \times d}$
10:                $\phi_{N+1}^{(A)}(h) = h + f(h\theta_{\text{down}}^{(A)})\theta_{\text{up}}^{(A)}$
11:        **end if**
12:    **end for**
13:    Sample batch $(x, y) \sim \mathcal{D}_t$
14:    Compute loss $\mathcal{L}$
15:    Update all $\phi^{(A)}$
16:    Append each $F_A(k)$ to $\mathcal{F}_{\text{hist}}$
17: **end for**

---

## B   ALGORITHM FOR ADAPTER GATING

To support inference with multiple adapters per task, TIDE uses a Fisher-weighted gating mechanism. Algorithm 2 outlines this process. Fisher traces determine adapter weights, which are normalised and used to compute a weighted combination of adapter outputs. This ensures that adapters which historically contributed more to preserving knowledge receive proportionally higher influence during inference. The gating mechanism generalises hard selection into a continuous mixture model, enabling robustness against varying levels of forgetting.

---

**Algorithm 2** Fisher-Weighted Adapter Gating

---

**Require:** Input $\mathbf{x}_t$, adapters $\{\phi_i^{(A)}\}_{i=1}^N$, Fisher traces $\{\mathcal{F}_i\}_{i=1}^N$
**Ensure:** Output prediction $\mathbf{y}_t$
1: $Z \leftarrow \sum_{j=1}^N \mathcal{F}_j$
2: **for** $i = 1$ to $N$ **do**
3:     $W_i \leftarrow \mathcal{F}_i/Z$ {Normalize Fisher traces}
4: **end for**
5: **for** $i = 1$ to $N$ **do**
6:     $\mathbf{h}_i \leftarrow \phi_i^{(A)}(\mathbf{x}_t)$ {Computer adapter outputs.}
7: **end for**
8: $\mathbf{y}_t \leftarrow \sum_{i=1}^N W_i \cdot \mathbf{h}_i$ {Compute weighted sum.}
9: **return** $\mathbf{y}_t$

---

## C    COMPLEXITY ANALYSIS.

Let $T$ be the number of tasks, $L$ the number of adapter-inserted layers, $d$ the model dimension at those layers, and $r$ the adapter bottleneck rank. Under the standard bottleneck adapter parameterisation (down-projection $d \to r$, up-projection $r \to d$), the worst-case parameter overhead from expansion is

$$\mathcal{O}\big(T \cdot L \cdot d \cdot r\big),$$

which we conservatively summarise as $\mathcal{O}(T \cdot L \cdot d^2)$ when $r = \Theta(d)$. In practice, $r \ll d$, so actual growth is much smaller. If we cap adapters per task to $M$, the bound becomes $\mathcal{O}(T \cdot M \cdot L \cdot d \cdot r)$.

Fisher estimation (we use a diagonal/trace approximation) requires one backwards pass over a minibatch of size $B$ when it is computed; thus, a single Fisher update costs roughly the same as one training update and can be rare (e.g., only at expansion checks), making its per-step cost negligible. The forgetting score itself is $\mathcal{O}(n)$ in the number of stored parameters as they are vector operations over stored $\theta^{*(A)}$ and precomputed Fisher diagonals.

At inference, adapter outputs are combined by a weighted sum. If a task has $N$ adapters active, inference cost increases by the cost of $N$ adapter forward passes at the chosen layers plus a cheap linear combination; therefore, inference FLOPs scale with $N \cdot L \cdot d \cdot r$. In our experiments $N$ remains small (cap $M \leq 10$), so the additional runtime is modest. Overall, TIDE trades a bounded amount of extra computation for targeted growth instead of complete model duplication. We include GPU memory and parameter count in Appendix L to quantify this trade-off.

## D    FISHER TRACE ESTIMATION

To decide when expansion is required, our method monitors the Fisher information trace during training. This section provides additional details on how the trace is estimated at each batch. Given a model with parameters $\theta$ and a batch of data $(x, y)$ drawn from the training stream, we estimate the diagonal Fisher information as

$$F(\theta) \approx \frac{1}{|B|} \sum_{(x,y) \in B} \nabla_\theta \log p_\theta(y \mid x) \odot \nabla_\theta \log p_\theta(y \mid x),$$

where $\odot$ denotes elementwise multiplication. In practice, the gradient of the log-likelihood with respect to $\theta$ is computed once per batch using automatic differentiation. We do not accumulate full diagonals; instead, we compute the scalar trace by summing over parameters:

$$\mathrm{Tr}(F) = \sum_i F_{ii}.$$

At each training step, a forward pass computes $\log p_\theta(y \mid x)$ for the current batch. A backward pass with the loss set to the negative log-likelihood accumulates gradients with respect to all parameters. The squared gradients are summed across all parameters to yield the per-batch trace. This scalar is normalised by the batch size to account for variability in gradient magnitude across tasks.

Let $F_t$ denote the Fisher-weighted forgetting signal computed after task $t$. To stabilise the expansion trigger and manage noise in forgetting, we maintain a moving average

$$F_{\mathrm{hist}}^{(t)} = (1 - \gamma) \, F_{\mathrm{hist}}^{(t-1)} \, + \, \gamma \, F_t,$$

where $\gamma \in (0, 1)$ is a smoothing coefficient.

## E    IMPACT OF PARAMETER EXPANSION ON FORGETTING

**Lemma 1.** *Let the model parameters be decomposed as $w = (\theta, \phi)$, where $\theta$ are the pre-expansion parameters and $\phi$ are the newly introduced adapter parameters. Let $\ell_A(\theta, \phi)$ be the loss of a previously learned task A, and let $F_A$ denote its FIM at the expansion point $(\theta_A^\star, \phi_0)$. Assume that the Fisher matrix is block diagonal in these coordinates:*

$$F_A \; = \; \begin{bmatrix} F_{\theta\theta} & 0 \\ 0 & 0 \end{bmatrix}.$$

*Define the Fisher-weighted second-order approximation of the change in the loss of task A under an update $\Delta w = (\Delta\theta, \Delta\phi)$ by*

$$\Delta\ell_A^{\mathrm{F}}(\Delta w) \; := \; \frac{1}{2}\,\Delta w^\top F_A \Delta w.$$

*Then for any update that is confined to the expanded subspace with $\Delta\theta = 0$, we have*

$$\Delta\ell_A^{\mathrm{F}}(0, \Delta\phi) = 0.$$

*In particular, within the Fisher approximation, updates restricted to $\phi$ do not increase the loss of task A.*

*Proof.* Substitute $\Delta w = (0, \Delta\phi)$ into the quadratic form:

$$\Delta\ell_A^{\mathrm{F}}(0, \Delta\phi) = \frac{1}{2} \begin{bmatrix} 0^\top & \Delta\phi^\top \end{bmatrix} \begin{bmatrix} F_{\theta\theta} & 0 \\ 0 & 0 \end{bmatrix} \begin{bmatrix} 0 \\ \Delta\phi \end{bmatrix}.$$

$$\Delta\ell_A^{\mathrm{F}}(0, \Delta\phi) = \frac{1}{2}\,\Delta\phi^\top 0\,\Delta\phi = 0.$$

Thus any update confined to the adapter parameters incurs zero Fisher-weighted forgetting on task $A$ at the expansion point. $\qquad\square$

## F  PLASTICITY DIRECTION WITH BOUNDED FORGETTING

**Lemma 2.** *Let B be a new task with loss $\ell_B(\theta, \phi)$, and let*

$$g_B := \nabla_\phi \ell_B(\theta_A^\star, \phi_0)$$

*denote its gradient with respect to the adapter parameters at the expansion point. Let the Fisher matrix of task A satisfy*

$$F_A \; = \; \begin{bmatrix} F_{\theta\theta} & 0 \\ 0 & 0 \end{bmatrix},$$

*and define the Fisher-weighted forgetting measure as in Lemma 1:*

$$\Delta\ell_A^{\mathrm{F}}(0, \Delta\phi) = \frac{1}{2}(0, \Delta\phi)^\top F_A(0, \Delta\phi).$$

*Assume $g_B \neq 0$. Then for any tolerance $\varepsilon > 0$ there exists an update $\Delta\phi$ such that:*

*(i) $\langle g_B, \Delta\phi \rangle < 0$;*

*(ii) $\Delta\ell_A^{\mathrm{F}}(0, \Delta\phi) \leq \varepsilon$.*

*Under $F_A$, condition (ii) holds with equality 0 for every $\Delta\phi$.*

*Proof.* From Lemma 1 for any $\Delta\phi$,

$$\Delta\ell_A^{\mathrm{F}}(0, \Delta\phi) = 0,$$

so condition (ii) is automatically satisfied for all $\Delta\phi$, and in particular for any prescribed $\varepsilon > 0$.

It remains to satisfy (i). Since $g_B \neq 0$, choose

$$\Delta\phi := -\eta g_B, \qquad \eta > 0.$$

Then

$$\langle g_B, \Delta\phi \rangle = \langle g_B, -\eta g_B \rangle = -\eta\|g_B\|^2 < 0,$$

so the update decreases $\ell_B$ to first order. Because $\Delta\ell_A^{\mathrm{F}}(0, \Delta\phi) = 0$, both (i) and (ii) hold simultaneously. $\qquad\square$

**Remark 1.** *If instead the Fisher block for $\phi$ is positive semidefinite, i.e.*

$$F_A = \begin{bmatrix} F_{\theta\theta} & 0 \\ 0 & F_{\phi\phi} \end{bmatrix}, \qquad F_{\phi\phi} \succeq 0,$$

*then choosing $\Delta\phi = -\eta g_B$ with sufficiently small $\eta > 0$ yields*

$$\Delta\ell_A^{\mathrm{F}}(0, \Delta\phi) = \frac{1}{2}\,\eta^2\,g_B^\top F_{\phi\phi}\,g_B \leq \varepsilon$$

*for any tolerance $\varepsilon > 0$, while still ensuring $\langle g_B, \Delta\phi \rangle < 0$. Thus the existence of a beneficial direction with bounded forgetting holds under any finite curvature in the adapter subspace; the case $F_{\phi\phi} = 0$ simply yields zero forgetting for all $\Delta\phi$.*

# G    PROOF FOR THEOREM 1

*Proof.* Let $e_i = \hat{\theta}_i - \theta$ with biases $b_i = \mathbb{E}[e_i]$ and cross-covariances

$$\Omega_{ij} = \text{Cov}(e_i, e_j) = \mathbb{E}\big[(e_i - b_i)(e_j - b_j)^\top\big], \qquad i, j \in \{1, 2\}.$$

Fix a symmetric positive definite matrix $W \in \mathbb{R}^{d \times d}$ and define the interpolation

$$\hat{\theta}(\alpha) = \alpha\hat{\theta}_1 + (1 - \alpha)\hat{\theta}_2, \qquad e(\alpha) = \hat{\theta}(\alpha) - \theta = \alpha e_1 + (1 - \alpha)e_2.$$

We study the risk

$$R(\alpha) = \mathbb{E}\big[\|e(\alpha)\|_W^2\big] = \mathbb{E}\big[e(\alpha)^\top W e(\alpha)\big].$$

Using bilinearity,

$$\begin{aligned}
e(\alpha)^\top W e(\alpha) &= (\alpha e_1 + (1 - \alpha)e_2)^\top W(\alpha e_1 + (1 - \alpha)e_2) \\
&= \alpha^2 e_1^\top W e_1 + (1 - \alpha)^2 e_2^\top W e_2 + \alpha(1 - \alpha)\big(e_1^\top W e_2 + e_2^\top W e_1\big).
\end{aligned}$$

Taking expectations and using linearity of $\mathbb{E}[\cdot]$,

$$R(\alpha) = \alpha^2 T_{11} + (1 - \alpha)^2 T_{22} + \alpha(1 - \alpha)\big(T_{12} + T_{21}\big), \tag{3}$$

where we abbreviate the scalars

$$T_{ij} := \mathbb{E}[e_i^\top W e_j], \qquad i, j \in \{1, 2\}.$$

Because $W = W^\top$ and each $T_{ij}$ is a scalar,

$$e_2^\top W e_1 = (e_1^\top W^\top e_2)^\top = (e_1^\top W e_2)^\top = e_1^\top W e_2,$$

hence $T_{21} = T_{12}$. Substituting into equation 3 yields

$$R(\alpha) = \alpha^2 T_{11} + (1 - \alpha)^2 T_{22} + 2\alpha(1 - \alpha)T_{12}. \tag{4}$$

Write the centered errors $\tilde{e}_i := e_i - b_i$ so that $\mathbb{E}[\tilde{e}_i] = 0$. Then

$$\begin{aligned}
\mathbb{E}[e_i^\top W e_j] &= \mathbb{E}[(\tilde{e}_i + b_i)^\top W(\tilde{e}_j + b_j)] \\
&= \mathbb{E}[\tilde{e}_i^\top W \tilde{e}_j] + \mathbb{E}[\tilde{e}_i^\top W b_j] + \mathbb{E}[b_i^\top W \tilde{e}_j] + b_i^\top W b_j.
\end{aligned}$$

Since $\mathbb{E}[\tilde{e}_i] = 0$, the mixed terms vanish: $\mathbb{E}[\tilde{e}_i^\top W b_j] = \mathbb{E}[\tilde{e}_i]^\top W b_j = 0$ and $\mathbb{E}[b_i^\top W \tilde{e}_j] = b_i^\top W \mathbb{E}[\tilde{e}_j] = 0$. Thus

$$\mathbb{E}[e_i^\top W e_j] = \mathbb{E}[\tilde{e}_i^\top W \tilde{e}_j] + b_i^\top W b_j. \tag{5}$$

For the first term, use the trace identity $x^\top A y = \text{tr}(A y x^\top)$:

$$\mathbb{E}[\tilde{e}_i^\top W \tilde{e}_j] = \mathbb{E}\big[\text{tr}(W \tilde{e}_j \tilde{e}_i^\top)\big] = \text{tr}\big(W \mathbb{E}[\tilde{e}_j \tilde{e}_i^\top]\big).$$

Noting $\mathbb{E}[\tilde{e}_j \tilde{e}_i^\top] = \Omega_{ji}$ (by the definition of cross-covariance),

$$\mathbb{E}[\tilde{e}_i^\top W \tilde{e}_j] = \text{tr}(W \Omega_{ji}).$$

Moreover, $\Omega_{ji} = \Omega_{ij}^\top$ and $W = W^\top$ imply

$$\text{tr}(W \Omega_{ji}) = \text{tr}(W \Omega_{ij}^\top) = \text{tr}\big((W \Omega_{ij}^\top)^\top\big) = \text{tr}(\Omega_{ij} W^\top) = \text{tr}(\Omega_{ij} W) = \text{tr}(W \Omega_{ij}).$$

Combining with equation 5,

$$T_{ij} = \mathbb{E}[e_i^\top W e_j] = \text{tr}(W \Omega_{ij}) + b_i^\top W b_j. \tag{6}$$

Let $U_{ij} := \text{tr}(W \Omega_{ij})$ and $B_{ij} := b_i^\top W b_j$. Then $T_{ij} = U_{ij} + B_{ij}$ and equation 4 becomes

$$R(\alpha) = \alpha^2(U_{11} + B_{11}) + (1 - \alpha)^2(U_{22} + B_{22}) + 2\alpha(1 - \alpha)(U_{12} + B_{12}).$$

Expand $(1 - \alpha)^2 = 1 - 2\alpha + \alpha^2$ and $2\alpha(1 - \alpha) = 2\alpha - 2\alpha^2$:

$$R(\alpha) = \alpha^2(U_{11} + B_{11}) + (1 - 2\alpha + \alpha^2)(U_{22} + B_{22}) + (2\alpha - 2\alpha^2)(U_{12} + B_{12})$$

$$= \underbrace{\left[U_{11} + U_{22} - 2U_{12} + B_{11} + B_{22} - 2B_{12}\right]}_{=:A} \alpha^2$$

$$+ 2\underbrace{\left[U_{12} - U_{22} + B_{12} - B_{22}\right]}_{=:B} \alpha + \underbrace{(U_{22} + B_{22})}_{=:C}.$$

Hence we have the quadratic form

$$R(\alpha) = A\alpha^2 + B\alpha + C. \tag{7}$$

Note $B_{11} + B_{22} - 2B_{12} = (b_1 - b_2)^\top W(b_1 - b_2)$. Also,

$$U_{11} + U_{22} - 2U_{12} = \operatorname{tr}\left(W(\Omega_{11} + \Omega_{22} - 2\Omega_{12})\right).$$

Using $U_{12} = \operatorname{tr}(W\Omega_{12}) = \operatorname{tr}(W\Omega_{21})$ (since $\Omega_{21} = \Omega_{12}^\top$ and $W = W^\top$), we can write

$$U_{11} + U_{22} - 2U_{12} = \operatorname{tr}\left(W(\Omega_{11} + \Omega_{22} - \Omega_{12} - \Omega_{21})\right).$$

Therefore

$$A = \operatorname{tr}\left(W(\Omega_{11} + \Omega_{22} - \Omega_{12} - \Omega_{21})\right) + (b_1 - b_2)^\top W(b_1 - b_2) =: D. \tag{8}$$

Similarly,

$$B = 2\left[(U_{12} + B_{12}) - (U_{22} + B_{22})\right] = 2\left(T_{12} - T_{22}\right)$$

$$= 2\left(\operatorname{tr}(W(\Omega_{12} - \Omega_{22})) + b_1^\top W b_2 - b_2^\top W b_2\right),$$

and

$$C = U_{22} + B_{22} = \operatorname{tr}(W\Omega_{22}) + b_2^\top W b_2.$$

These coincide with the expressions in the theorem statement.

Now we show strict convexity and uniqueness of the minimiser. Differentiate equation 7:

$$R'(\alpha) = 2A\alpha + B, \qquad R''(\alpha) = 2A.$$

By equation 8 and the assumption $D > 0$, we have $A > 0$, hence $R''(\alpha) > 0$ for all $\alpha$ and $R$ is strictly convex with a unique stationary point. Solving $R'(\alpha) = 0$ gives the unconstrained minimizer

$$\alpha^\star = -\frac{B}{2A}.$$

Using the expression for $B$,

$$-\frac{B}{2} = \operatorname{tr}\left(W(\Omega_{22} - \Omega_{12})\right) + b_2^\top W(b_2 - b_1),$$

so

$$\alpha^\star = \frac{\operatorname{tr}\left(W(\Omega_{22} - \Omega_{12})\right) + b_2^\top W(b_2 - b_1)}{D}. \tag{9}$$

Because $R$ is strictly convex, the constrained minimizer on $[0, 1]$ is the Euclidean projection

$$\alpha_{\mathrm{opt}} = \Pi_{[0,1]}(\alpha^\star) = \min\{1, \max\{0, \alpha^\star\}\}.$$

Define the scalar

$$Q := \mathbb{E}\left[(e_1 - e_2)^\top W(e_1 - e_2)\right] = T_{11} + T_{22} - T_{12} - T_{21}.$$

Using $T_{21} = T_{12}$ and the expressions for $T_{ij}$,

$$Q = \operatorname{tr}\left(W(\Omega_{11} + \Omega_{22} - \Omega_{12} - \Omega_{21})\right) + (b_1 - b_2)^\top W(b_1 - b_2) = A.$$

Since $W \succ 0$, $Q \geq 0$ with equality iff $e_1 - e_2 = 0$ a.s. in the $W$-norm; the assumption $D = A > 0$ rules out this degenerate case and guarantees strict convexity.

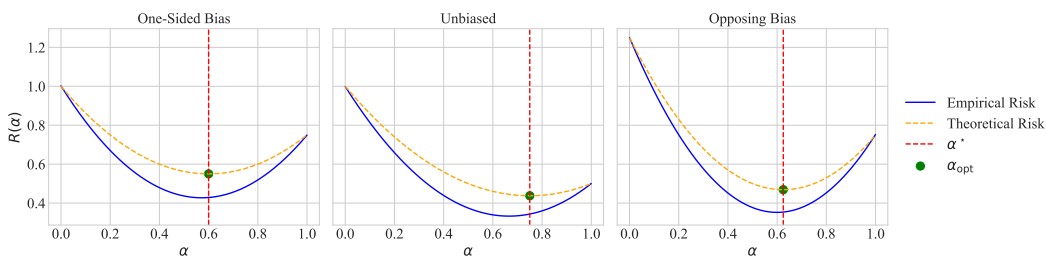

Figure 4: Empirical validation of Theorem 1. Risk $R(\alpha)$ as a function of the interpolation weight $\alpha$, with three bias configurations. Blue lines indicate Monte Carlo estimates, while orange dashed lines show the theoretical quadratic form. The unconstrained minimizer $\alpha^\star$ (red dashed) and the constrained optimum $\alpha_{\mathrm{opt}}$ (green circle) confirm both convexity and the closed-form solution.

Let $d = 1$, $W = 1$, $b_i \in \mathbb{R}$, variances $\omega_i^2 := \Omega_{ii}$, covariance $\omega_{12} := \Omega_{12} = \Omega_{21}$. Then

$$U_{ij} = \mathrm{tr}(W\Omega_{ij}) = \Omega_{ij}, \qquad B_{ij} = b_i b_j.$$

Hence

$$A = \left(\omega_1^2 + \omega_2^2 - 2\omega_{12}\right) + (b_1 - b_2)^2, \quad B = 2\left(\omega_{12} - \omega_2^2 + b_1 b_2 - b_2^2\right) = 2\left((\omega_{12} - \omega_2^2) - b_2(b_2 - b_1)\right).$$

Therefore

$$-\frac{B}{2} = (\omega_2^2 - \omega_{12}) + b_2(b_2 - b_1), \qquad \alpha^\star = \frac{\omega_2^2 - \omega_{12} + b_2(b_2 - b_1)}{\omega_1^2 + \omega_2^2 - 2\omega_{12} + (b_1 - b_2)^2},$$

and $\alpha_{\mathrm{opt}} = \Pi_{[0,1]}(\alpha^\star)$. If, in addition, $b_1 = b_2 = 0$, then

$$\alpha^\star = \frac{\omega_2^2 - \omega_{12}}{\omega_1^2 + \omega_2^2 - 2\omega_{12}}, \qquad \text{and if } \omega_{12} = 0, \ \alpha^\star = \frac{\omega_2^2}{\omega_1^2 + \omega_2^2}.$$

All identities follow from substituting $U_{ij} = \Omega_{ij}$ and $B_{ij} = b_i b_j$ into the general formulas above. $\qquad\square$

## H  EMPIRCAL VALIDATION OF THEOREM 1

To validate the theoretical results of Theorem 1, we performed controlled experiments examining the risk $R(\alpha)$ as a function of the interpolation parameter $\alpha$. We considered estimators of dimension $d = 5$ and generated $10^5$ Monte Carlo samples for each $\alpha \in [0, 1]$. The covariance matrices $\Omega_{ij}$ were chosen with moderate correlations to ensure that both variance and covariance terms contributed to the risk. Three different bias settings were tested:

$$b_1 = b_2 = 0, \quad b_1 = (0.5, 0, \ldots, 0)^\top, \ b_2 = 0, \quad b_1 = (0.5, 0, \ldots, 0)^\top, \ b_2 = (-0.5, 0, \ldots, 0)^\top,$$

corresponding respectively to the unbiased case, a one-sided bias, and opposing biases.

For each configuration, we constructed $\hat{\theta}(\alpha) = \alpha\hat{\theta}_1 + (1 - \alpha)\hat{\theta}_2$ and computed the empirical risk as the average squared $W$-norm error across samples, with $W = I$ for simplicity. In parallel, we evaluated the theoretical quadratic form of $R(\alpha)$ from Theorem 1.

Figure 4 compares empirical and theoretical risks under the three bias settings. In all cases, the risk curves are strictly convex in $\alpha$, consistent with the theorem. The unconstrained minimizer $\alpha^\star$ and the constrained optimum $\alpha_{\mathrm{opt}}$ closely match the empirical minima of the Monte Carlo curves. These results confirm both the convexity of $R(\alpha)$ and the accuracy of the closed-form expression for the optimal interpolation weight. With dimension $d = 1$, $W = [1]$, $b_1 = 0$ and $b_2 = \sqrt{\frac{2}{3}}$, we get $\alpha = 0.4$.

## I  PROOF FOR THEOREM 2

*Proof.* For each task $t \in \{1, \ldots, T\}$, assume a pointwise upper bound

$$\mathcal{R}_t(m) \leq g_t(m) := a_t e^{-\gamma m} + s\, m^2, \qquad a_t > 0, \ \gamma > 0, \ s > 0, \ m \geq 0.$$

Define the average plasticity scale $A := \frac{1}{T} \sum_{t=1}^{T} a_t > 0$ and the *envelope*

$$R(M) := Ae^{-\gamma M} + s M^2, \qquad M \geq 0.$$

First, fix $t$. Differentiate $g_t$:

$$g_t'(m) = -\gamma a_t e^{-\gamma m} + 2s\, m, \qquad g_t''(m) = \gamma^2 a_t e^{-\gamma m} + 2s.$$

Because $a_t, \gamma, s > 0$ and $e^{-\gamma m} > 0$ for all $m$, we have $g_t''(m) > 0$ for all $m \geq 0$. Thus $g_t$ is strictly convex on $[0, \infty)$ and therefore admits a unique (unconstrained) minimiser $m_t^\star$ solving $g_t'(m_t^\star) = 0$.

The same differentiation for $R$ gives

$$R'(M) = -\gamma Ae^{-\gamma M} + 2s\, M, \qquad R''(M) = \gamma^2 Ae^{-\gamma M} + 2s > 0 \quad \forall M \geq 0,$$

so $R$ is strictly convex on $[0, \infty)$ with a unique minimizer $M^\star$ solving $R'(M^\star) = 0$.

Set $R'(M) = 0$:

$$-\gamma Ae^{-\gamma M} + 2s\, M = 0 \iff 2s\, M = \gamma Ae^{-\gamma M} \iff 2s\, M\, e^{\gamma M} = \gamma A.$$

Introduce $x := \gamma M$ (so $M = x/\gamma$). Then

$$2s \frac{x}{\gamma} e^x = \gamma A \iff xe^x = \frac{\gamma^2 A}{2s}.$$

By definition of the principal branch of the Lambert $W$ function (the inverse of $x \mapsto xe^x$ on $[0, \infty)$),

$$x = W\left(\frac{\gamma^2 A}{2s}\right) \implies M^\star = \frac{1}{\gamma} W\left(\frac{\gamma^2 A}{2s}\right).$$

The argument $\frac{\gamma^2 A}{2s} > 0$ ensures $W$ is real and unique on the principal branch, matching the uniqueness implied by strict convexity.

For a cap $M \geq 0$, consider minimizing $g_t$ over $[0, M]$. Since $g_t$ is strictly convex with unique unconstrained minimiser $m_t^\star$, the constrained minimiser is

$$m_t(M) := \arg \min_{m \in [0,M]} g_t(m) = \begin{cases} m_t^\star, & \text{if } m_t^\star \leq M, \\ M, & \text{if } m_t^\star > M. \end{cases}$$

Equivalently, $m_t(M) = \min\{m_t^\star, M\}$.

We now compare $g_t\big(m_t(M)\big)$ with $g_t(M)$:

- If $m_t^\star \leq M$, then by optimality of $m_t^\star$ we have $g_t(m_t^\star) \leq g_t(M)$, hence $g_t\big(m_t(M)\big) = g_t(m_t^\star) \leq g_t(M)$.

- If $m_t^\star > M$, then $m_t(M) = M$ and $g_t\big(m_t(M)\big) = g_t(M)$.

In all cases,

$$g_t\big(m_t(M)\big) \leq g_t(M) = a_t e^{-\gamma M} + s\, M^2. \tag{10}$$

Since $\mathcal{R}_t(\cdot) \leq g_t(\cdot)$ pointwise,

$$\mathcal{R}_t\big(m_t(M)\big) \leq g_t\big(m_t(M)\big) \leq a_t e^{-\gamma M} + s\, M^2.$$

Averaging over tasks and recalling $A = \frac{1}{T} \sum_t a_t$,

$$\bar{\mathcal{R}}(M) := \frac{1}{T} \sum_{t=1}^{T} \mathcal{R}_t\big(m_t(M)\big) \leq \frac{1}{T} \sum_{t=1}^{T} \left(a_t e^{-\gamma M} + s\, M^2\right)$$

$$= \left(\frac{1}{T} \sum_{t=1}^{T} a_t\right) e^{-\gamma M} + s\, M^2 = Ae^{-\gamma M} + s\, M^2 = R(M).$$

Thus $R$ is an upper envelope for the average capped risk, and minimizing $R$ yields the cap that minimizes this envelope.

Suppose the admissible caps are integers $M \in \{0, 1, \ldots, M_{\max}\}$. Because $R$ is strictly convex and continuous on $[0, \infty)$ with a unique minimizer $M^\star$, the restriction of $R$ to the integer grid $\{0, 1, \ldots, M_{\max}\}$ attains its minimum at one of the two nearest integers to $M^\star$ (or at the boundary if $M^\star \notin [0, M_{\max}]$). Formally, let

$$\lfloor M^\star \rfloor \leq M^\star \leq \lceil M^\star \rceil, \qquad \mathcal{I} := \{0, 1, \ldots, M_{\max}\}.$$

Then

$$M_{\mathrm{opt}} = \arg\min_{m \in \mathcal{I}} R(m) \in \left\{ \Pi_{\mathcal{I}}\big(\lfloor M^\star \rfloor\big), \ \Pi_{\mathcal{I}}\big(\lceil M^\star \rceil\big) \right\},$$

where $\Pi_{\mathcal{I}}$ denotes Euclidean projection onto the discrete set $\mathcal{I}$ (i.e., clipping to $[0, M_{\max}]$ and then choosing the nearest integer). Equivalently, $M_{\mathrm{opt}}$ is the projection of $M^\star$ onto $\{0, 1, \ldots, M_{\max}\}$:

$$M_{\mathrm{opt}} = \arg\min_{m \in \{0, \ldots, M_{\max}\}} |m - M^\star|.$$

In the tie case $M^\star = k + \frac{1}{2}$ with $k \in \mathbb{Z}$, both $k$ and $k+1$ attain the same value of $R$ by symmetry of strict convexity about $M^\star$, and either choice is optimal (the projection is set-valued there). $\qquad\square$

## J EMPIRICAL VALIDATION OF THEOREM 2

We empirically examine the trade-off between plasticity and stability in adapter-based models using the risk function

$$R(M) = Ae^{-\gamma M} + sM^2,$$

as described in Theorem 2, where $M$ denotes the number of adapters. Here, $A > 0$ represents the plasticity scale, $\gamma > 0$ the diminishing-returns rate, and $s > 0$ the stability scale. For each setting, we evaluate $R(M)$ over integer adapter caps $M \in \{0, \ldots, 15\}$ with $\gamma = 0.3$ and compare with the theoretical continuous minimizer $M^\star = \frac{1}{\gamma} W\left(\frac{\gamma^2 A}{2s}\right)$, where $W$ is the Lambert $W$-function.

Two experimental scenarios are considered:

1. **Varying plasticity $A$ with fixed stability $s$**: This setup illustrates how increasing the plasticity scale shifts the exponential term in $R(M)$, changing both the location and depth of the minimum risk.

2. **Varying stability $s$ with fixed plasticity $A$**: Here, we explore how increasing the stability scale strengthens the quadratic penalty, showing the diminishing returns of adding more adapters.

Results in Figure 5 confirm that the projected Lambert $W$ prediction closely aligns with the integer-optimal cap across a variety of settings, illustrating the convexity of $R(M)$ and the practical utility of the closed-form solution. We also find that a higher plasticity scale aligns with a higher adapter cap, while a higher stability scale aligns with a lower adapter cap. Our experiments use $A = 10.0$, $s = 0.05$ for $M = 10$.

## K DATASETS

We provide additional details for the datasets used in our experiments. Table 5 shows the statistics of each image dataset. Each dataset is split into a sequence of class-incremental tasks, with 5 classes introduced per task for CIFAR-100 and CORe50, 10 per task for CUB, and 20 per task for IN-R, IN-A and IN-1k. All images are resized to $224 \times 224$ and normalised to match the input distribution expected by the pre-trained transformer backbone. Each model is evaluated in a class-incremental setting without task labels at test time.

**CIFAR-100.** CIFAR-100 contains 60,000 colour images, each of size $32 \times 32$, evenly distributed across 100 classes (600 images per class). The dataset is relatively low-resolution compared to the others, which often leads to different feature statistics once resized to $224 \times 224$ for transformer-based models.

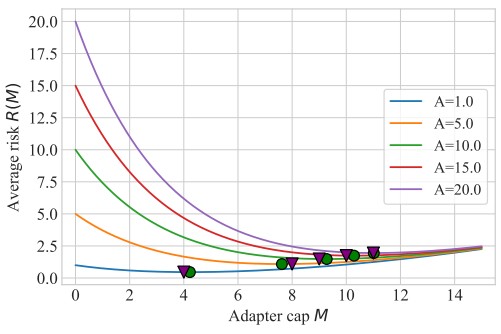 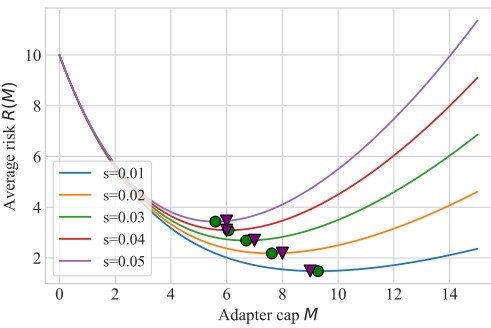

(a) Varying plasticity $A$ with fixed stability $s$.  (b) Varying stability $s$ with fixed plasticity $A$.

Figure 5: Adapter cap with risk curves for varying stability and plasticity scales. Lines indicate the continuous risk $R(M)$, green circles mark the projected theoretical minimiser $M^\star$, and purple triangles indicate the empirically optimal integer adapter cap $M_{\mathrm{opt}}$. Both subfigures confirm that the Lambert $W$ prediction accurately identifies near-optimal adapter caps.

**CUB-200-2011.** CUB provides 11,788 images over 200 bird species. Images are collected from the web and vary in resolution, typically around $200 \times 200$ pixels, though sizes are not uniform. Each image includes fine-grained annotations (part locations, bounding boxes, and segmentations), but only class labels are used here. The small number of examples per class makes this dataset particularly sensitive to overfitting under incremental training.

**ImageNet-R (IN-R).** ImageNet-Rendition consists of 30,000 images covering 200 ImageNet classes. Unlike the original ImageNet, these are non-naturalistic renditions (e.g., paintings, cartoons, embroidery). Image resolutions vary, often considerably larger than $224 \times 224$, and must be downsampled. The dataset is explicitly designed to evaluate robustness to domain shift.

**ImageNet-A (IN-A).** ImageNet-A contains 7,500 images from 200 classes selected from ImageNet. The images are natural photographs that were adversarially curated: only samples consistently misclassified by standard ImageNet-trained classifiers are retained. Image resolutions are unconstrained and typically high. The small dataset size combined with its adversarial nature makes it challenging for incremental learners.

**ImageNet-1k (IN-1k).** The ILSVRC-2012 ImageNet benchmark comprises $\sim$1.28 million training images and 50,000 validation images across 1,000 classes. The raw images are high-resolution (up to $512 \times 512$ or greater), with highly varied aspect ratios. This dataset represents the standard large-scale benchmark and provides a contrasting setting compared to the smaller CIL benchmarks.

**CORe50.** CORe50 includes 50 household objects captured across 11 recording sessions, with variations in lighting, background, and context. It contains 164,866 images, each originally $128 \times 128$. The dataset is video-based, with sequences of objects being manipulated; individual frames are extracted to form the benchmark. The strong correlations between frames introduce additional challenges for rehearsal-based methods.

**VQAv2.** The VQAv2 dataset contains $\sim$1.1 million questions paired with images from the MS-COCO dataset, with human-provided answers. Each question is associated with 10 answers from annotators, making it suitable for evaluating both accuracy and robustness to linguistic ambiguity. The image set spans both the COCO train and validation splits (2014 release), with images loaded in their original resolution and aspect ratio. For continual learning evaluation, we use the binary Yes/No subset, which is a widely studied and computationally tractable partition of the full dataset. Preprocessing within the loader is limited to normalising answers to lowercase Yes/No labels, while image resizing and normalisation are deferred to downstream transforms.

**ScienceQA-IMG.** The ScienceQA-IMG dataset extends the ScienceQA benchmark with $\sim$21k multiple-choice science questions, where a subset is grounded in diagrammatic or photographic images. Each sample consists of a question, a set of candidate answers, an optional supporting rationale, and an image when available. Images are loaded in their raw resolution without resizing inside the dataset class, consistent with other loaders. Any preprocessing, such as resizing (e.g., to $224 \times 224$) or normalisation, is applied externally using task-specific transforms. Questions and

choices are returned as raw strings, with text tokenisation handled at the model input stage. This benchmark evaluates not only visual grounding but also the ability to integrate textual reasoning and multimodal information under continual learning constraints.

Table 5: Image dataset statistics. All images are resized to $224 \times 224$ for training, but the default image sizes and number of examples are reported here.

| Dataset | Classes | Examples | Default Image Size |
|---|---|---|---|
| CIFAR-100 | 100 | 60,000 | $32 \times 32$ |
| CUB | 200 | 11,788 | Varying, $\sim 200 \times 200$ |
| ImageNet-R | 200 | 30,000 | Varying, up to high-res |
| ImageNet-A | 200 | 7,500 | Varying, up to high-res |
| ImageNet-1k | 1,000 | 1.28M | Varying, $\leq 512 \times 512$ |
| CORe50 | 50 | 164,866 | $128 \times 128$ |

## L  COMPLEXITY

Table 6 reports the number of additional parameters introduced by each method, alongside the corresponding average accuracy $\mathcal{A}_B$. The results reveal clear trade-offs between efficiency and predictive performance. L2P achieves the smallest footprint ($< 0.2$M parameters), and is constant across training steps. DualPrompt increases the parameter budget to approximately 1M, improving accuracy but still lagging behind more adaptive strategies. CODA-Prompt incurs the highest overhead ($\approx 4$M parameters), yet its accuracy gains are inconsistent, particularly on IN-A. By contrast, SEMA and TIDE maintain a compact parameter budget (0.5–0.8M) while achieving accuracies comparable to or exceeding larger models, demonstrating the benefits of dynamic expansion. EASE sits in between, trading slightly higher parameter cost ($\approx 1.3$M) for consistently strong accuracy.

Table 6: Number of added parameters (P) used in model deployment (in Millions) and average accuracy $\mathcal{A}_B$ across benchmarks.

| Method | CIFAR-100 | | CUB | | IN-R | | IN-A | | IN-1k | | CORe50 | |
|---|---|---|---|---|---|---|---|---|---|---|---|---|
| | P | $\mathcal{A}_B \uparrow$ | P | $\mathcal{A}_B \uparrow$ | P | $\mathcal{A}_B \uparrow$ | P | $\mathcal{A}_B \uparrow$ | P | $\mathcal{A}_B \uparrow$ | P | $\mathcal{A}_B \uparrow$ |
| L2P | 0.12 | 77.87 | 0.15 | 66.25 | 0.20 | 52.92 | 0.20 | 41.71 | 0.20 | 52.50 | 0.11 | 64.69 |
| DP | 1.02 | 81.15 | 1.05 | 70.34 | 1.09 | 58.74 | 1.09 | 47.29 | 1.09 | 57.89 | 0.98 | 65.92 |
| CODA-P | 3.91 | 81.96 | 3.95 | 73.37 | 3.99 | 55.08 | 3.99 | 35.02 | 3.99 | 58.79 | 3.87 | 68.08 |
| EASE | 1.18 | 85.80 | 1.18 | 86.81 | 1.51 | 70.58 | 1.32 | 55.04 | 1.30 | 83.13 | 1.18 | 80.85 |
| SEMA | 0.64 | **86.01** | 0.66 | 87.51 | 0.61 | 71.19 | 0.56 | 53.45 | 0.55 | 83.04 | 0.55 | 82.41 |
| TIDE | 0.512 | 85.41 | 0.62 | **87.56** | 0.85 | **72.61** | 1.25 | **61.00** | 1.19 | **83.16** | 1.05 | **84.58** |

Table 8 presents the estimated training compute cost per iteration in TFLOPs. Since backbone computation dominates, absolute values remain similar across methods, but relative differences are still informative. L2P is the most compute-efficient ($< 0.1$ TFLOPs), reflecting its fixed and lightweight prompt design. DualPrompt and EASE fall in the mid-range (0.4–0.7 TFLOPs), consistent with their moderately sized parameter expansions. CODA-Prompt is the most computationally demanding ($\approx 1.6$ TFLOPs), reflecting the overhead of maintaining large prompt sets. Notably, both SEMA and TIDE achieve strong accuracy with compute demands similar to or below DualPrompt, confirming that limited adaptive expansion can reduce overall training costs.

Table 7: Training times (s/it) for CIFAR-100 and IN-A.

| Methods | CIFAR-100 | IN-A |
|---|---|---|
| SEDEM | 21.62s | 11.51s |
| MoE-Adapters | 23.43s | 10.63s |
| EASE | **21.49**s | **10.01**s |
| SEMA | 22.58s | 11.73s |
| TIDE | 22.83s | 12.30s |

Table 7 compares the average computational cost of TIDE against several SOTA continual learning baselines in terms of per-iteration training time on CIFAR-100 and IN-A. While TIDE elevates parameter effectiveness, this comes at a higher computational expense. The increased cost is primarily due to TIDE's need to calculate a forgetting score at initial iterations for a new task. We also provide curves for the progression of these statistics as tasks are learned in Figure 6.

Table 8: Training compute cost (TFLOPs) across benchmarks. Values scaled by dataset size relative to CIFAR-100.

| Method | CIFAR-100 | CUB | IN-R | IN-A | IN-1k | CORe50 |
|--------|-----------|-----|------|------|-------|--------|
| L2P | **196.10** | **232.54** | **117.66** | **118.83** | **117.89** | **137.36** |
| DP | 301.49 | 365.14 | 180.89 | 178.85 | 180.35 | 440.15 |
| CODA-P | 351.95 | 421.22 | 211.17 | 211.58 | 211.14 | 382.33 |
| EASE | 324.27 | 383.88 | 194.56 | 190.83 | 191.28 | 150.16 |
| SEMA | 301.55 | 360.16 | 180.93 | 185.83 | 183.52 | 140.60 |
| TIDE | 251.84 | 300.19 | 151.10 | 187.78 | 149.25 | 161.86 |

Table 9 quantifies the memory footprint of additional parameters, measured in MiB under standard 32-bit storage. The trends mirror the parameter counts in Table 6. L2P again achieves the smallest footprint ($< 1$ MiB), while CODA-Prompt is the most memory-intensive ($\approx 16$ MiB). DualPrompt and EASE lie in the intermediate range ($\approx 4$–$6$ MiB), and SEMA consistently stays below 3 MiB. TIDE exhibits slightly higher memory than SEMA (2–5 MiB), but remains far more efficient than CODA-Prompt while achieving comparable or superior accuracy.

Table 9: Memory usage (MiB) of additional parameters across benchmarks.

| Method | CIFAR-100 | CUB | IN-R | IN-A | IN-1k | CORe50 |
|--------|-----------|-----|------|------|-------|--------|
| L2P | **0.49** | **0.60** | **0.80** | **0.80** | **0.80** | **0.44** |
| DP | 4.09 | 4.20 | 4.39 | 4.39 | 4.38 | 3.93 |
| CODA-P | 15.7 | 15.8 | 16.0 | 16.0 | 16.0 | 15.5 |
| EASE | 4.75 | 4.75 | 6.06 | 5.31 | 5.23 | 4.75 |
| SEMA | 2.58 | 2.64 | 2.47 | 2.24 | 2.21 | 2.22 |
| TIDE | 2.05 | 2.50 | 3.41 | 5.00 | 4.78 | 4.20 |

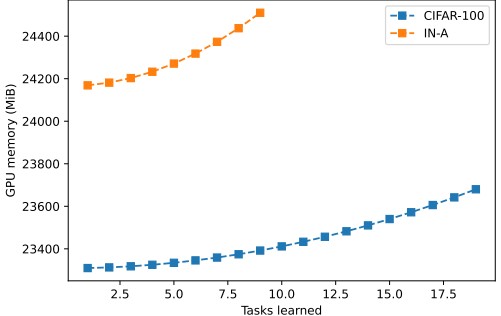
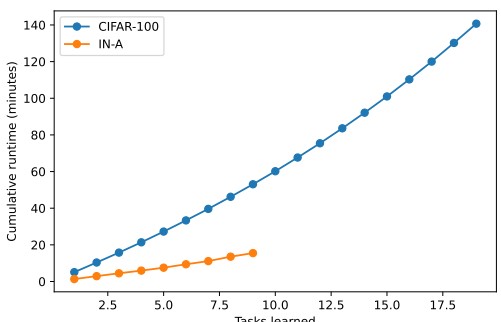

(a) Memory growth as tasks are learned.      (b) Cumulative runtime as tasks are learned.

Figure 6: Memory and runtime curves as more tasks are learned for CIFAR-100 and IN-A.

## M   ADAPTER USAGE

To contextualise TIDE within existing dynamically expandable architectures, we summarise in Table 10 the criteria used by different methods to decide when to introduce new capacity. While prior approaches typically allocate adapters at task boundaries or according to fixed or heuristic schedules, TIDE differs by linking expansion directly to statistically significant forgetting measured during training. This distinction allows TIDE to respond to interference as it emerges, enabling fine-grained and task-sensitive growth rather than relying on externally imposed expansion points.

Figure 9 reports the accuracy obtained by each adapter across tasks. Several important patterns emerge. First, usage is not uniform: while some adapters are consistently assigned to multiple tasks with high accuracy (e.g., Adapter 1 for early tasks and Adapter 4 for mid-sequence tasks), others play a more specialised role, showing strong performance only in a subset of tasks. This suggests that

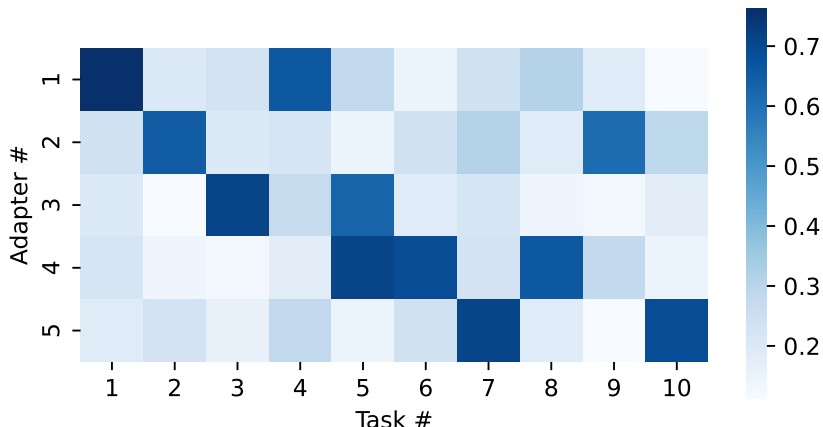

Figure 7: Adapter usage heatmap showing task-wise accuracies for each assigned adapter. Darker colours correspond to higher accuracy.

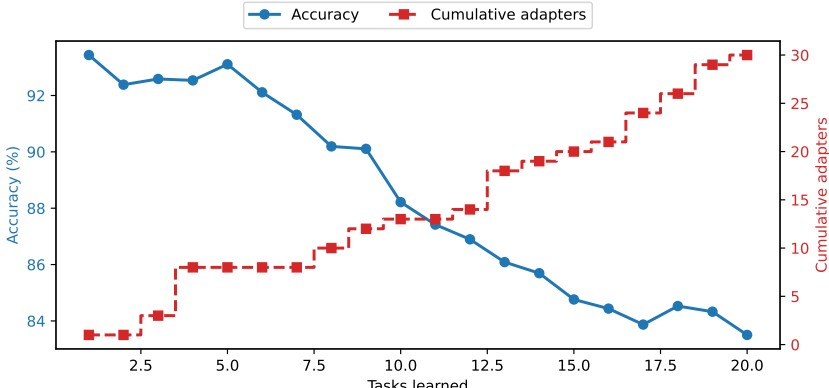

Figure 8: Timeline of accuracy and adapter expansion in terms of tasks learned on CIFAR-100.

adapters are capturing complementary representational subspaces rather than redundantly encoding the same information.

Figure 8 shows the cumulative number of adapters added during training on CIFAR-100 compared to the accuracy curve during training on a task sequence. We observe a correlation between new adapters and sustained accuracy.

Second, the distribution of adapter usage highlights a balance between reuse and specialisation. For example, Adapter 2 is reused across dissimilar tasks with moderate accuracy, whereas Adapter 5 demonstrates selective but strong accuracy in later tasks. This indicates that the method is capable of exploiting shared representations when beneficial, while still allocating dedicated capacity when tasks diverge. Such behaviour is consistent with the underlying principle of adaptive expansion: avoid unnecessary growth while ensuring robustness against interference.

Finally, the heatmap illustrates the TIDE's ability to prevent catastrophic forgetting through flexible routing. High accuracies across most task–adapter assignments demonstrate that previously allocated adapters remain reliable, even as new adapters are introduced. This validates the expansion strategy's effectiveness in preserving knowledge retention while supporting continual adaptation.

Figure 9 illustrates two training dynamics under different adapter allocation strategies. On the left, we observe the standard TIDE process, where adapters are introduced incrementally in response to detected forgetting events. This adaptive expansion leads to consistent improvements in training accuracy following each adapter addition. Notably, the points of intervention (marked by the vertical green lines) align with inflection points in the accuracy curve, suggesting that targeted expansion successfully mitigates performance degradation and stabilises learning as new capacity is allocated

Table 10: Comparison of expansion triggers between TIDE and prior adapter-based or dynamically expandable methods. TIDE differs by using a statistically grounded, Fisher-weighted forgetting criterion rather than heuristic or boundary-based decisions.

| Method | Expansion Trigger |
|---|---|
| **TIDE (ours)** | Expands only when a past task exhibits *statistically significant forgetting*, detected via Fisher-weighted forgetting scores compared against historical distributions (see Equation 1). |
| **MoE Adapters** (Yu et al., 2024) | Adds new experts at *task boundaries*. Expansion is coupled to the introduction of each new task; no intra-task or dynamic trigger is used. |
| **EASE** (Zhou et al., 2024) | Expansion triggered once per task based on *task-level distributional shift* and predefined heuristics determining when additional subspaces are needed. |
| **SEMA** (Wang et al., 2025) | Performs *self-expansion at task boundaries* by estimating task difficulty and uncertainty; growth is limited to a single decision per task rather than continuous monitoring. |
| **Task Expansion** | Always expands at each new task, adding a fixed number of adapters unconditionally. |
| **Even Expansion** | Uniformly distributes expansions across each task at pre-chosen intervals, regardless of observed forgetting. |
| **Random Expansion** | Expands adapters at random training steps using a uniform schedule, ignoring task difficulty or interference. |

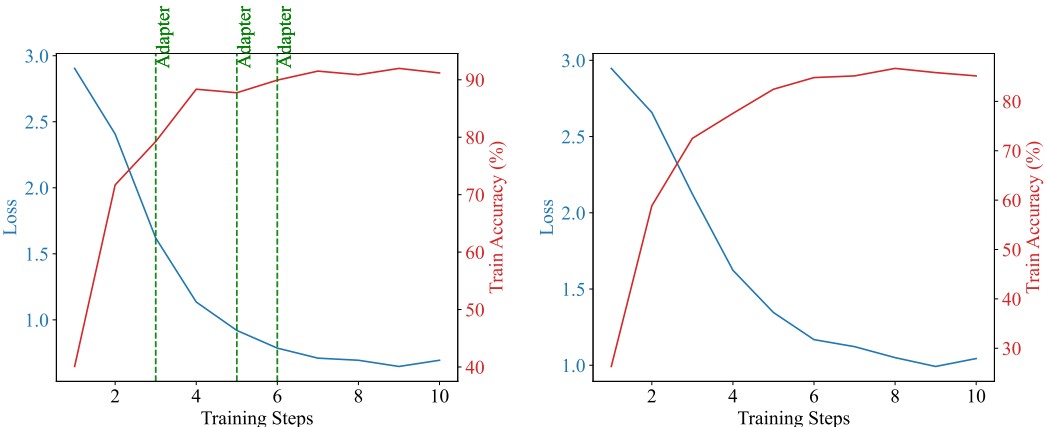

Figure 9: Training progress on a task from ImageNet-A. The left plot shows the normal TIDE process as adapters are added when forgetting is detected, and the right plot shows the same task when five adapters are initialised prior to training with evenly weighted gating.

only when necessary. In contrast, the right panel depicts the same task under a naive strategy where five adapters are initialised prior to training with equal gating weights. Here, accuracy increases smoothly over time without the sharp stepwise improvements observed in the adaptive case. However, the overall accuracy plateau is lower, and the loss remains consistently higher compared to the adaptive strategy. This indicates that while over-parameterisation from the outset avoids forgetting-triggered disruptions, it dilutes the learning signal across redundant modules and leads to inefficient use of capacity.

Figure 10 shows the forgetting curves before and after adapter expansion, demonstrating the flattening of the forgetting statistics after an adapter is added. This indicates that adapters do succeed in mitigating the Fisher-based forgetting score.

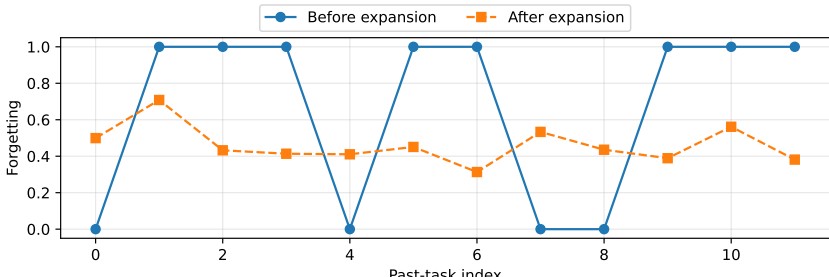

Figure 10: Per-task normalised forgetting curves showing Fisher-weighted forgetting before and after each expansion trigger. The blue curve shows the normalised forgetting value at the moment the expansion test is evaluated. Due to per-task normalisation, tasks whose first forgetting spike is small relative to later values appear at 0. The orange curve shows the post-expansion forgetting trajectory for each task.

## N    BUFFER SELECTION STRATEGIES

To evaluate the sensitivity of TIDE to different buffer selection strategies, we conducted experiments comparing random sampling, reservoir sampling, and recency bias. The results in Figure 11 demonstrate that reservoir sampling consistently yields higher accuracy across datasets, indicating more effective retention of representative samples over time. Random sampling performs moderately, while recency bias shows lower accuracy, suggesting that overemphasising recent samples can degrade long-term knowledge retention. This aligns with the intuition that continual learning requires balanced coverage of past tasks rather than overfitting to the latest distributions. By storing examples in proportion to their frequency, reservoir sampling maintains a more faithful approximation of the original task distribution, leading to greater robustness.

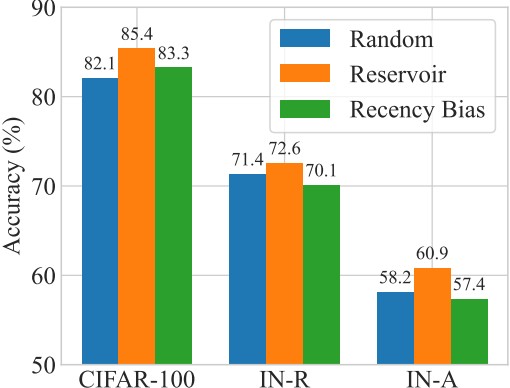

Figure 11: TIDE performance using different buffer selection strategies.

## O    ADAPTER EXPANSION STRATEGIES

Here, we provide details on expansion strategies in Table 3, which isolate the effect of when capacity is added during continual learning, while keeping how much capacity is added fixed across conditions. For each dataset, we specify an adapter budget per task (i.e., the total number of new adapter modules to be inserted across all adapter layers) and ensure all strategies insert exactly the same number of adapters per task and in the same layers. Unless otherwise noted, newly added adapters are trained alongside previously learned adapters and the backbone under identical optimisation and buffer settings. The only difference between strategies is the timing of insertions.

**Task Expansion.**    At each task boundary, the full per-task adapter budget is inserted in one shot before training on the new task begins. This mirrors common expansion baselines that allocate

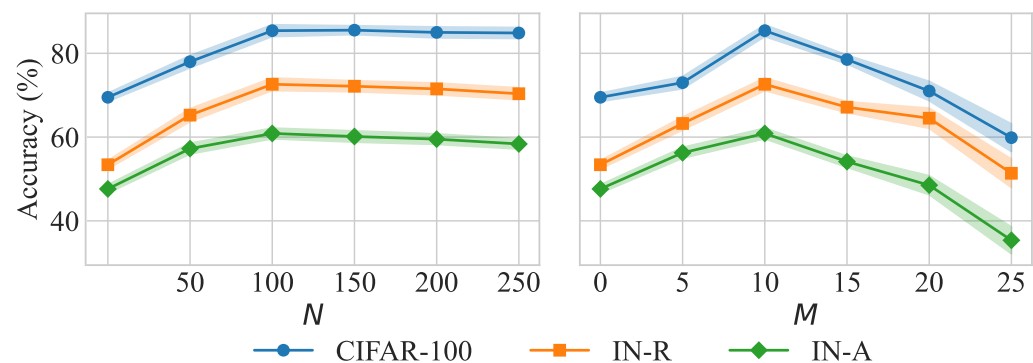

Figure 12: Sensitivity of TIDE to buffer size $N$ and adapter capacity $M$.

capacity only when a new task arrives. While this approach concentrates plasticity at the start of a task and avoids mid-epoch architectural changes, it can over-allocate capacity early if the task is simple, or under-allocate if interference emerges later in training.

**Random Expansion.**  Here the same per-task budget is distributed uniformly at random across the training steps of each task. Concretely, given $S$ optimization steps and a budget of $K$ insertions, we sample $K$ distinct steps $s_1, \ldots, s_K \sim \text{Uniform}\{1, \ldots, S\}$ without replacement, inserting one adapter at each sampled step. This ablates any inductive bias toward early or late expansion while preserving the total capacity. Because insertions may cluster, the schedule can remain uneven across runs.

**Even Expansion.**  In this strategy, the $K$ insertions are spaced at approximately even intervals over the $S$ steps of training, using $\lfloor i \cdot S/(K+1) \rfloor$ for $i = 1, \ldots, K$. This ensures plasticity is distributed smoothly across the task, providing early flexibility for adaptation while still leaving capacity for late-arising interference.

For all three strategies, the adapter architecture, initialisation, and gating are identical. The optimiser, schedule, and buffer policy are fixed, and the total number of adapters per task is matched. When an insertion occurs, the new adapter is immediately initialised and participates in training from that step onward.

Figure 13 highlights the effectiveness of TIDE by showing a correlation between the Fisher-weighted forgetting variance and the actual accuracy drops during training on a sequence of tasks in CIFAR-100.

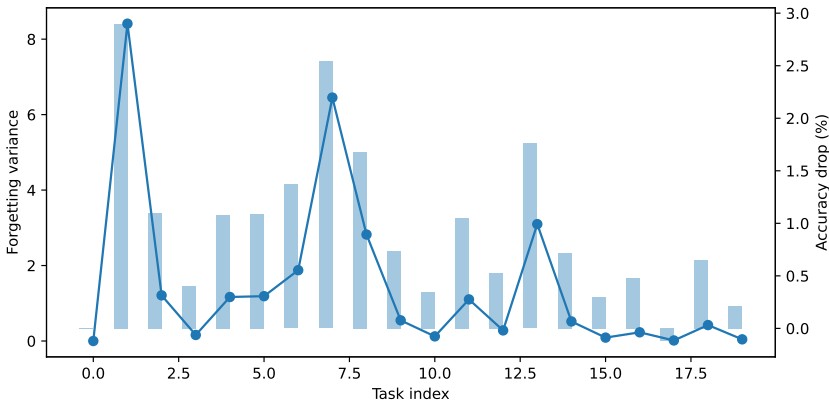

Figure 13: Variance of the Fisher-based forgetting signal and its correlation with accuracy drops in CIFAR-100.

## P    FURTHER SENSITIVITY

Figure 12 shows the sensitivity of TIDE to two key hyperparameters: the adapter cap $M$ to determine the maximum number of adapters, and the validation buffer size $N$. The validation buffer benefits from more examples as it more effectively represents the data stream for a given task; however, after 100 examples, we observe diminishing returns at the expense of greater memory complexity. For the adapter cap $M$, performance degrades if more than 10 adapters are allowed. Tasks that are difficult to classify may saturate the model with parameters, leading to decreased performance.

## Q    ETHICS STATEMENT

All experiments are conducted on publicly available datasets, and the proposed methodology does not introduce foreseeable risks of harm or misuse.

## R    LARGE LANGUAGE MODELS

Large Language Models (LLMs) were used as assistive tools to aid in polishing the writing and improving readability. They were not involved in research ideation, experimental design, or interpretation of results. The authors take full responsibility for the content of this paper.

