# OpenReview forum: "Timed Dynamic Expansion for Continual Learning"
_ICLR.cc/2026/Conference — Submitted to ICLR 2026_

### Official Review · Reviewer_7TJs · 2025-10-28

**Soundness:** 2
**Presentation:** 1
**Contribution:** 2
**Rating:** 4
**Confidence:** 3

**Summary:**

The authors propose an adaptive model expansion based continual learning method that creates adapters upon detecting forgetting. They also introduce a forgetting metric based on Fisher information and evaluate the approach on several benchmarks.

**Strengths:**

1. The authors improve expansion based methods by inserting adapters only when the current task exhibits forgetting, which prevents unnecessary growth.
2. The authors conduct experiments on several benchmarks and report improvements over competing methods.

**Weaknesses:**

The mathematical expressions are not rigorous, and many discussions rely heavily on strong assumptions:
1. It is unclear which dataset the loss in the equation at line 186 is evaluated on, $D_A$ or $D_t$? This must be clarified, because if the loss is evaluated on $D_t$, the definition of forgetting in Eq.(1) would be incorrect.
2. The authors assume small parameter perturbations, meaning the updates on new tasks are small, which naturally implies little or no forgetting. Moreover, if $||\Delta\theta||$ is small, the second-order term should be smaller than the first-order term. On what basis do you conclude that “the leading-order change in loss is driven by the second derivative”?
3. The assumption in lines 211–215 that both the first and second derivatives are zero is not generally observed in practice.
4. The statements of Theorems 1 and 2 are puzzling and do not appear to align with the claims the authors intend to make.

**Questions:**

Besides the weaknesses, we have the following questions:
1. In line 244, what does the notation F_hist represent?
2. In the experiments, why not report the model’s performance on mitigating forgetting using standard metrics, such as Backward Transfer, or the forgetting metric defined by the authors?

---

> ### Author Response · Authors · 2025-11-23
>
> We thank the reviewer for their constructive comments and for recognising the contributions of selective expansion. Below, we address each point in turn and clarify the mathematical assumptions and notation used in the paper.
>
> **W1.** In all theoretical developments, the loss ${L}$ in the Taylor expansion is the empirical risk on the reference task $A$, i.e.,
> $
> {L}_A(\theta) = {L}(\theta;{D}_A),
> $
> or its validation split ${D}_A^{\mathrm{val}}$ where specified. Since forgetting is defined as the increase of task-$A$ loss relative to $\theta^{*(A)}$, both Eq. (1) and the surrounding analysis are evaluated on ${D}_A$. We will explicitly write ${L}_A(\theta)$ at its first occurrence for clarity.
>
> **W2.** The Taylor expansion in line 186 is taken around the optimum $\theta^{\*(A)}$ of task $A$, where $\nabla_{\theta} L_A(\theta^{\*(A)}) \approx 0$. Thus, the first-order term satisfies
> $
> \|\nabla_{\theta} {L}_A(\theta^{*(A)})^\top \Delta\theta\| \le \varepsilon \|\Delta\theta\|,
> $
> with $\varepsilon$ small. In contrast, the quadratic term
> $
> \tfrac{1}{2}\Delta\theta^\top H_A \Delta\theta
> $
> can be significant even for small $\|\Delta\theta\|$ when updates align with directions of high curvature and thus large eigenvalues of $H_A$. This is the motivation behind the Fisher-weighted forgetting metric, where forgetting is dominated by movement along sensitive, high-Fisher directions, not by step size alone. We will clarify this reasoning immediately after Eq. 1.
>
> **W3.** The statement in lines 211-215 refers to the behaviour of newly added parameters at the moment of expansion. These parameters are introduced through a gated, zero-initialised residual adapter. At this initialisation:
> $
> \frac{\partial {L}_A}{\partial \phi_j} = 0,
> \qquad
> \frac{\partial^2 {L}_A}{\partial\theta_i \partial\phi_j} = 0,
> $
> because $\phi$ does not participate in the forward computation on ${D}_A$. This is a structural property of the architecture at init, not a global assumption about derivatives during training. We will clarify that this assumption applies only at initialisation and is standard in adapter-based designs.
>
> **W4.** We appreciate the request for clearer statements. Theorems 1 and 2 aim to formalise two claims:
> - (Theorem 1) Under the block-diagonal Fisher structure induced at adapter initialisation, updates flowing into the expanded subspace $\phi$ do not increase the Fisher-weighted forgetting measure. Therefore, expansion provides a feasible subspace for plasticity without harming task $A$.
> - (Theorem 2) There exists a feasible update direction in the expanded space that achieves progress on the new task while keeping the Fisher-weighted increase in loss on task $A$ below a fixed tolerance $\tau$.
>
> We will restate both theorems more precisely and move full derivations to the appendix, ensuring that assumptions and claims align transparently.
>
> **Q1.** $F_{\text{hist}}$ denotes a moving average of the Fisher-weighted forgetting signal. It stabilises the trigger by smoothing short-term noise and ensuring that expansion occurs only when forgetting persists. We will explicitly add the update rule in Appendix C:
>
> Let $F_t$ denote the Fisher-weighted forgetting signal computed after task $t$. To stabilise the expansion trigger, we maintain a moving average
> $F_{\text{hist}}^{(t)}
>     = (1 - \gamma) F_{\text{hist}}^{(t-1)}
>       + \gamma\, F_{t}
> $
> where $\gamma \in (0,1)$ is a smoothing coefficient.
>
> **Q2.** We agree that standard measures provide stronger insight. In the revised manuscript, we will report the Backward Transfer (BWT). We avoid reporting the proposed Fisher-based forgetting metric because doing so would unfairly favour our method, given that the metric is tightly coupled to the TIDE expansion mechanism and not a general forgetting measure.
>
> | Method | CIFAR-100 | CUB    | IN-R   | IN-A   | IN-1k  | CORe50 |
> |--------|-----------|--------|--------|--------|--------|--------|
> | L2P    | -15.98    | -15.59 | -13.52 | -16.56 | -9.01  | -9.64  |
> | DP     | -11.45    | -14.21 | -11.01 | -14.91 | -8.46  | -9.47  |
> | CODA-P | -11.39    | -12.86 | -12.50 | -20.42 | -8.41  | -9.01  |
> | EASE   | -7.65     | -6.98  | -7.67  | -10.40 | -1.26  | -6.45  |
> | SEMA   | -7.61     | -6.45  | -7.41  | -9.81  | -1.28  | -6.81  |
> | **TIDE** | **-7.49** | **-6.43** | **-7.32** | **-9.16** | **-1.25** | **-7.13** |

---

> ### Author Response · Authors · 2025-11-27
>
> We hope our responses have addressed your questions and clarified the contributions of our work. As the discussion period is nearing its end, we would kindly appreciate any further comments or guidance on remaining concerns. Thank you again for your time and constructive feedback.

---

> > ### Comment · Reviewer_7TJs · 2025-11-27
> >
> > We thank the authors for their detailed response and for effectively addressing most of our concerns.
> >
> > However, we remain unsatisfied with the response regarding W4. The descriptions of Theorem 1 and Theorem 2 provided in the rebuttal appear to be fundamentally different from the theorems currently presented in the manuscript. Rather than clarifying the existing theoretical framework, the response suggests entirely new formulations, which has only increased our confusion regarding the paper's theoretical grounding.
> >
> > Consequently, we have decided to maintain our current score. However, we remain open to discussion. If the authors can provide the explicit mathematical formulations and proofs for the "restated" Theorems 1 and 2 mentioned in the rebuttal, we will review them and consider raising our score.

---

> > > ### Author Response · Authors · 2025-11-28
> > >
> > > We thank the reviewer for highlighting these concerns, and apologise for the confusion in our last response. We had attempted to rephrase some ideas related to the theory in the paper, and hope to clarify matters by providing Lemmas 1 and 2 in the Appendix (which we mistakenly did not provide in our previous rebuttal), which relate to the intuition and forgetting formulations in our work, and by more clearly stating the intention of Theorems 1 and 2.
> > >
> > > First, we provide and prove two Lemmas:
> > >
> > > **Lemma 1.** We show, using the full Fisher quadratic form, that updates in $\phi$ incur zero second-order increase in the Fisher-weighted loss on task A. (i.e. expansion provides a subspace for plasticity without harming task A). (Appendix E)
> > >
> > > **Lemma 2.** We show that there are useful (improving) directions for the new task in the same subspace that keeps Fisher-weighted loss under a tolerance $\epsilon$. (Appendix F)
> > >
> > > We hope that these Lemmas reinforce the theory used to motivate our method. Theorems 1 and 2 aim to show that the forgetting threshold $\alpha$ and the adapter cap $M$ are not heuristic, but correspond to unique minimisers of convex risk functions. We clarify in the paper:
> > >
> > > - Theorem 1 shows that the risk associated with expansion decisions is a strictly convex function of $\alpha$, ensuring the existence of a single, well-defined optimal threshold.
> > >
> > > - Theorem 2 formalises a stability–plasticity trade-off by showing that the long-term risk is a strictly convex function of the adapter count, yielding a unique optimal cap on the number of adapters per task.
> > >
> > > We hope that this improves the clarity of our paper, and are happy to answer any further questions the reviewers may have.

---

### Official Review · Reviewer_BJSz · 2025-10-28

**Soundness:** 4
**Presentation:** 4
**Contribution:** 4
**Rating:** 6
**Confidence:** 4

**Summary:**

Continual learning is challenged by catastrophic forgetting: training on later tasks overwrites information learned from earlier tasks. To mitigate, existing methods expand the network for each new task, leading to a large growth in parameter counts. Authors propose a flexible expansion method that install small adapter modules into a pre-trained backbone when forgetting for an old task is measured. To measure forgetting, authors weight the change in parameter p to its reference value after training task t with the Fisher information matrix. If forgetting is statistically significant, a new adapter is integrated. At inference, adapter outputs are weighted by their FIM. This method is extensively evaluated in the class-incremental setting on 5 image and two mixed-domain datasets, where it outperforms existing CIL methods.

**Strengths:**

- Novelty of the idea: tying parameter expansion to fisher based forgetting
- Extensive set of experiments on also large scale ImageNet-scale datasets (with 25+ repetitions per dataset to average numbers)
- Several ablation studies integrated that analyse the impact of different componens (e.g., adapter $\alpha$)
- Presentation of the paper easy to follow (I enjoyed reading the paper)

**Weaknesses:**

- I kindly disagree with the notion that the adapters are inserted for an old task. Rather, I think that the adapters are inserted for the current task, to prevent it from overwriting the important parameters of the old task.

- Its unclear to me how/if adapters are trained: once an adapter is integrated for old task A, how will task A know of its existence? I mean, the newly-installed adapter was not there when the model was trained on task A? Then, if the adapter is trained (? line15 in algorithm 1 suggests so), how is it trained to not interfere with A? Shouldn't the old task A completely ignore the newly-installed adapter (because, again, it was not there during original trainin on task A)? Please clarify the role of

**Questions:**

- Is the backbone even trained? Figure 1 and Algorithm 1 seem to suggest that only the adapters are trained, and the rest is frozen? But, then, how can an adapter be trained for task 1, because in task 1 there is no forgetting (because its the first task, no historical performance is available yet), and hence no adapter needs to be installed for task 1?
- Whats the difference between $F^{A}\_{ij}$ and $F^{A}\_{i}$ (Why do you switch in equation 1?)
- line 315: is this really a prediction? or rather the hidden state, computed as the weighted average? Or, do you refer to the actual label prediction, but where the respective adapters inside the models have been weighted accordingly?
- line 306: why use the historical importance? Won't this become outdated as the adapter is trained?

(Most of my questions and concerns revolve around the way the adapters are used. Please clarify this in the discussion period)

---

> ### Author Response · Authors · 2025-11-23
>
> We thank the reviewer for their thoughtful comments and for highlighting both the strengths and weaknesses of the paper. Below, we address each of the weaknesses and answer the questions listed.
>
> **W1.** We agree that adapters created during training on task $t$ receive gradients only from the current task. Our naming convention (e.g., "adapter for task $A$'') refers to the task whose forgetting triggered the expansion, not to the dataset used to train the adapter. When task $A$ exhibits significant Fisher-weighted forgetting, TIDE allocates a new adapter to increase plasticity without modifying parameters that are important for $A$. Thus, the adapter is introduced because of $A$, but is trained using data from the current task $t$. Incidentally, this reinforces the need to maintain a memory buffer of past tasks, so that TIDE captures forgetting dynamics in prior tasks as well as the current task.
>
> **W2.** Adapters created due to forgetting on task $A$ begin with zero initial influence and zero Fisher block. This means that $\partial \mathcal{L}_A / \partial \phi_j = 0$ and $\partial^2 \mathcal{L}_A / (\partial \theta_i \partial \phi_j) = 0$ at initialization, yielding a guaranteed ``safe'' subspace for absorbing plasticity. The adapter need not be trained on task $A$; rather, it prevents harmful movement of $\theta$ along high-curvature directions for $A$. During inference, TIDE's Fisher-based gating assigns nonzero weight to an adapter only if it consistently represents low-interference directions, allowing old tasks to exploit expanded capacity without requiring retroactive training.
>
> **Q1.** The backbone is never trained during our experiments, and only adapters receive gradient updates. For the first task, TIDE instantiates an initial adapter before training begins; since no forgetting can occur for task $1$, no additional expansion is triggered. After the first adapter is instantiated prior to training, other adapters are added according to the TIDE process. We will make this detail clearer in the paper by specifying: "One adapter is instantiated prior to training, and all subsequent adapters are created by the TIDE process"
>
> **Q2.** We thank the reviewer for bringing this oversight to our attention. While $F_{ij}^{(A)}$ denotes items from the complete Fisher Information Matrix, TIDE uses a trace approximation $F_{i}^{(A)}$ due to its decreased computational cost and similar trends noted in Section 2. We have made changes to explain the shift in notation in Equation 1: "$F_{i}^{(A)}$ denotes the trace approximation of the Fisher score".
>
> **Q3.** The quantity in line 315 denotes the final adapter-combined representation used to compute the prediction. Each adapter produces a residual transformation, and these are aggregated via Fisher-based weights $W_i$. We will clarify this as the "final adapter-weighted representation for prediction."
>
> **Q4.** The historical importance $F_{\text{hist}}$ is an exponential moving average of Fisher diagonals. This stabilises gating, prevents oscillatory routing, and ensures that an adapter retains influence only if it consistently captures important directions over time. It is therefore not a static quantity and does not become outdated as training proceeds.

---

> > ### Comment · Reviewer_BJSz · 2025-11-24
> >
> > I thank the authors for their explanations. As noted in my initial review, my main concerns revolved around the naming of adapters and how and adapter **for an old task** can be trained, as that task's data is no longer available. This has been clarified in the authors' response; the adapter is inserted **in response to** an old task and adds new parameters (ie capacity) to learn new tasks without interfering with old parameters. I trust the authors to integrate this distinction (ie, their answers to W1 and W2) into their manuscript, possibly around line 157. Further, the new experimental results provided in response to the other reviewers' questions add to the paper. I am increasing my score.

---

### Official Review · Reviewer_4Jvj · 2025-10-31

**Soundness:** 3
**Presentation:** 4
**Contribution:** 3
**Rating:** 4
**Confidence:** 5

**Summary:**

This paper proposes TIDE, a continual learning framework that dynamically expands adapter modules only when statistically significant forgetting is detected. Instead of expanding at fixed task boundaries, TIDE continuously monitors Fisher-weighted forgetting scores and triggers adapter creation as needed. A Fisher-informed gating mechanism then aggregates multiple adapters per task during inference. The method aims to balance stability and plasticity by coupling capacity growth directly to observed forgetting, achieving strong performance across several CIL benchmarks and a MLLM setting with lower parameter overhead.

**Strengths:**

1. The paper is well-written, with clear illustrations and easy to follow
2. Introduces a novel forgetting-triggered expansion mechanism that adapts the capacity of the model during continual learning.
3. Achieves state-of-the-art or comparable accuracy on multiple benchmarks with fewer trainable parameters.
4. Extends beyond image classification to MLLM settings, showing method versatility.

**Weaknesses:**

1. Thanks for the efforts in reproducing all previous rehearsal-free methods using a replay buffer. While all compared methods are rehearsal-free, TIDE still relies on memory buffers for forgetting detection, raising concerns about its feasibility in a strictly rehearsal-free setting.

2. The approach closely resembles prior self-expansion or dynamic adapter methods.

3. The Fisher-based forgetting metric, based on trace approximation and small replay buffers, may be noisy. Its robustness to estimation errors is unclear, and reducing the memory buffer significantly hurts performance (Fig. 9).

4. Continual evaluation of forgetting for all past tasks could become costly as task numbers grow.

5. Visualizations showing when adapters expand during the whole CIL task sequences would clarify TIDE’s dynamics.

6. In Table 3, parameter counts for CORe50 between Task Expansion and TIDE do not differ much, weakening the claim of lower growth.

7. Since adapters are added when forgetting occurs, per-task forgetting curves before and after expansion are essential to demonstrate effectiveness.

**Questions:**

see weakness

---

> ### Author Response · Authors · 2025-11-23
>
> We thank the reviewers for their thorough and constructive feedback. We appreciate the positive remarks, and below, we address each concern point by point.
>
> 1. We agree that the reliance on memory buffers deserves clarification. The small buffer in TIDE is not used for rehearsal or retraining, but solely for statistical estimation of Fisher Information required to compute forgetting scores. It stores only minimal sufficient statistics and does not participate in gradient updates. To clarify this distinction, we will explicitly note in Section 3.2 that TIDE remains rehearsal-free in regard to parameter updates: “The mini-batches are not used to train the model, avoiding the extra complexity of full rehearsal methods”. We also consider that the memory buffer could be replaced with a buffer of feature-space prototypes, which mitigate forgetting with lower memory cost (Prototype-Guided Memory Replay for
> Continual Learning, IEEE Transactions on Neural Networks and Learning Systems 2023)
>
> 2.  We acknowledge that TIDE builds upon the adapter-based expansion paradigm. However, its contribution is orthogonal: TIDE introduces a statistically grounded trigger for expansion, not a new module type. Prior methods expand periodically or by heuristics, whereas TIDE links expansion directly to Fisher-weighted forgetting, providing an interpretable criterion we believe is applicable to other dynamic architectures.
> To highlight this distinction, we add a comparison table in Appendix K (Table 10), which shows the expansion triggers used by TIDE and prior work.
>
> 3. We appreciate this observation and agree that trace-based Fisher approximations can introduce stochastic variance. In practice, TIDE mitigates this through exponential averaging of Fisher estimates across mini-batches. To demonstrate robustness, we include an additional plot showing the variance of the Fisher-based forgetting signal and its correlation with actual accuracy drops (Figure 13).
>
> 4. We note that TIDE’s Fisher updates are incremental and diagonalised, so there is no recomputation of full matrices. The cost per iteration is $O(n)$ (Appendix C), comparable to a gradient step. We agree that this cost is expected to increase as more tasks are introduced, although it is offset by the decrease in trained parameters. This result is empirically observed in Appendix J, particularly on the ImageNet-1k dataset, which contains a diverse selection of 1000 classes. We find that the FLOPs decreases by up to 29.3\% compared to SOTA dynamic expansion approaches. We also include efficiency curves reporting runtime and memory growth as the number of tasks increases (Figure 6).
>
> 5. We appreciate the suggestion and include a timeline visualisation showing when new adapters are created during a training sequence. We also provide an adapter usage heatmap (Figure 7), illustrating how gating weights evolve as new adapters are integrated.
>
> 6. We agree that the parameter counts on CORe50 are similar across expansion strategies. CORe50 consists of highly heterogeneous object instances and substantial viewpoint and lighting variation, which leads TIDE to allocate capacity more frequently. In such settings, increased growth is expected and reflects genuine task dissimilarity. We find that while TIDE does not reduce parameters on every individual dataset, but allocates capacity when interference arises. This makes TIDE robust across a wide range of continual learning regimes. For closely related tasks, it suppresses unnecessary expansion (as observed on CIFAR-100 and CUB), whereas for more diverse or non-stationary streams like CORe50, it remains responsive to genuine forgetting and preserves performance by adding the needed adapters. This highlights the versatility of the framework, as TIDE increases parameters only when required for stability, and avoids redundant growth when tasks are similar.
>
> 7. We appreciate this request and include per-task forgetting curves showing Fisher-weighted forgetting before and after each expansion trigger (Figure 10).

---

> > ### Comment · Reviewer_4Jvj · 2025-11-23
> >
> > I thank the authors for their response. However, my concerns regarding the novelty of the expansion mechanism, the use of the replay buffer for Fisher information, and the forgetting metrics remain unresolved.
> >
> > I also have a follow-up question regarding the new Figure 10, which is currently confusing: Why does the forgetting metric range from 0 to 1? Additionally, why is the 'before expansion' forgetting exactly 0 for tasks 0, 4, 7, and 8?

---

> > > ### Author Response · Authors · 2025-11-26
> > >
> > > We thank the reviewer for their follow-up and for re-examining the updated material. We address the remaining concerns regarding 1. novelty, 2. the use of the buffer for Fisher estimation, 3. accounting for noise in the forgetting values, and 4. interpretation of the forgetting plots in Figure 10.
> > >
> > > **1.** While TIDE builds on the general idea of adapter-based expansion, our contributions are distinct from prior self-expansion methods in the mechanism that triggers growth. Existing approaches typically expand at fixed task boundaries or rely on heuristic rules tied to dataset segmentation, while TIDE introduces a time-sensitive approach based around forgetting that continuously monitors interference and allocates new adapters only when past tasks exhibit measurable degradation.
> > >
> > > **2.** TIDE uses a small buffer only to estimate Fisher-weighted forgetting, not for rehearsal or gradient replay. However, we appreciate the concern that maintaining such a buffer incurs extra memory expense. We align TIDE’s use of a memory buffer with SOTA continual learning approaches (STAR: Stability-Inducing Weight Perturbation for Continual Learning, ICLR 2025, Prioritized Generative Replay, ICLR 2025, Layerwise Proximal Replay: A Proximal Point Method for Online Continual Learning, ICML 2024) which similarly use a memory buffer to allow for rehearsal or validation on previously seen tasks.
> > >
> > > **3.** We acknowledge that trace–based Fisher estimates can be noisy. Therefore we take steps to ensure that TIDE is robust against noise. Expansion is not decided from a single forgetting value; instead, for each task $A$, we maintain a running mean of past forgetting scores $F_{\text{hist}}$. We add details and clarifications on this in Appendix D.
> > >
> > > Expansion occurs only when $p < \alpha$ for these averaged and batch-wise computation, so random fluctuations caused by trace noise are highly unlikely to trigger expansion. Moreover, the forgetting score itself averages curvature-weighted parameter shifts over many dimensions,
> > > $
> > > F_A(t) = \sum_i \mathcal{F}^{(A)}_i (\theta^{*(A)}_i - \theta^{(t)}_i)^2,
> > > $
> > > which further suppresses high-frequency noise. In practice, expansions align with sustained
> > > forgetting trends rather than isolated stochastic spikes, indicating that the statistical test
> > > successfully filters out noise.
> > >
> > > **4.** The forgetting axis is normalised to $[0,1]$ for visual comparability across datasets with  different Fisher magnitudes. In the tasks that have "0" forgetting, expansion occurs immediately following the first statistically significant forgetting event, and the plotted "before expansion" value corresponds to the forgetting score at the moment the trigger fires. Because we normalise per task, the first observed forgetting value is mapped to 0. This does not mean that raw forgetting is zero; rather, it reflects that for these tasks the initial forgetting spike is small relative to later oscillations. We will adjust the caption to make this explicit.

---

> ### Author Response · Authors · 2025-11-28
>
> We hope we have helped to clarify the contributions of our work. As the discussion period draws to a close, we would greatly appreciate any additional comments or guidance on any remaining concerns. We again thank the reviewers for their time and constructive feedback.

---

### Author Response · Authors · 2025-12-02

Dear Area Chair and Reviewers,

We thank the reviewers for their thoughtful engagement with our work, which presents a novel framework for model expansion based on the forgetting detected on previous tasks, and were encouraged by the feedback throughout the discussion. We appreciate the constructive comments, which has helped to strengthen our manuscript.

Our paper received **initial scores of 4,6,4**. We engaged in detailed discussion with all reviewers and amended our manuscript according to their insights. As a result, two reviewers indicated their intention to raise their scores. Reviewer BJSz **updated their score from 6 to 8** on November 24th, before scores were locked, while Reviewer 7TJs indicated that they would be willing to raise their score following a theoretical clarification which we provided. Unfortunately, we did not receive a response before discussion was closed. We strongly believe that had an incident not occurred during the rebuttal phase, our final ratings would have reached **4, 8, 6**. We summarise the changes made to our manuscript during the rebuttal process and how we addressed each reviewer’s main concerns.

**Reviewer 4Jvj.** We addressed the reviewer’s concerns regarding the cost of forgetting computations, provided visualisations of adapter expansions, the difference in parameter counts between datasets and provided per-task forgetting curves. Before the shortened discussion period, we further addressed concerns about the use of a memory buffer by comparing with SOTA continual learning methods and included details by which noise is reduced in the forgetting metric. We thank the reviewer for their feedback and hope that we satisfactorily addressed their remaining concerns.

**Reviewer BJSz.** We amended the manuscript with details on adapter initialisation and how forgetting in prior tasks is monitored and compensated. Following our rebuttal, **the reviewer increased their score from a 6 to an 8**. We thank the reviewer for their feedback during the discussion process.

**Reviewer 7TJs.** We corrected some inconsistencies in notation and clarified important theoretical points in our manuscript. Though the reviewer noted that we addressed most of their concerns, they noted some confusion regarding stated theory that could be further clarified, and **commented that if explicit mathematical formulations could be provided, they would consider raising their score**: "If the authors can provide the explicit mathematical formulations and proofs for the "restated" Theorems 1 and 2 mentioned in the rebuttal, we will review them and consider raising our score.". Prior to the end of discussion, **we included two further Lemmas on pages 14-15 in our manuscript to strengthen their concerns** about the theory provided in the paper. We strongly feel that the additions would have resolved these concerns, and thank the reviewer for their thoughtful discussion.

We **kindly ask that improvements to our manuscript are considered while making the final decision**. We again thank all reviewers for their comments, and thank the Area Chair for their consideration.

---

### Meta-Review · Area_Chair_cknh · 2025-12-12

**Summary:**

Reviewers generally acknowledged the potential of the proposed TIDE framework but raised a set of recurring concerns, particularly around novelty, theoretical clarity, and practical assumptions. Several reviewers questioned how distinct the method really is from prior dynamic expansion or adapter-based continual learning approaches, noting that while the trigger mechanism differs, the overall paradigm closely resembles existing self-expansion methods. A major concern was the reliance on a memory buffer to estimate Fisher-based forgetting - although the method is described as rehearsal-free, reviewers argued that this buffer weakens the claim and introduces feasibility, scalability, and noise issues, especially as task numbers grow. The robustness and interpretability of the Fisher-weighted forgetting metric were also questioned, including its sensitivity to approximation noise, normalization choices, and the computational cost of continually evaluating forgetting across all past tasks. On the theoretical side, one reviewer found the mathematical formulation insufficiently rigorous, with strong or unclear assumptions (e.g., small perturbations, zero derivatives at expansion) and confusion about where losses were evaluated.

**Reviewer Concerns:**

The rebuttal partially addressed concerns about the Fisher-weighted forgetting metric. The authors clarified how forgetting is computed, specified that losses are evaluated on the reference task, and explained the role of normalization in the plots. They also argued that noise is mitigated through exponential averaging and trend-based triggering rather than single measurements, and provided additional figures to illustrate variance, per-task forgetting, and expansion timing. Practical concerns about computational cost were also partly addressed by arguing that Fisher updates are incremental and comparable to a gradient step, with added efficiency curves to suggest manageable overhead.

However, several main concerns remain unresolved, among them novelty remains to be a core issue, as the rebuttal reframes TIDE’s contribution as a different expansion trigger but does not fully address the impression that the method is an incremental variant of prior adapter-based dynamic expansion approaches. Moreover, the performance improvement over prior work seems to be incremental as well on most benchmarks. The memory buffer remains another important open concern - although the authors clarify it is not used for rehearsal, its necessity continues to weaken the “rehearsal-free” claim and raises unanswered questions about scalability and robustness as task counts increase.

**Reviewer Scores:**

Reviewer 4Jvj (score 4) raised concerns about limited novelty relative to prior dynamic expansion methods, reliance on a memory buffer despite the “rehearsal-free” framing, noise and robustness of the Fisher-based forgetting metric, scalability of continual forgetting evaluation, and limited parameter savings on some datasets. Although the rebuttal addressed several practical issues—adding visualizations, per-task forgetting curves, and efficiency analyses—the reviewer explicitly stated that their core concerns regarding novelty, buffer reliance, and forgetting metrics remained unresolved. As a result, it is likely that this reviewer would have maintained the initial score.

Reviewer BJSz (score 6) raised concerns about conceptual clarity, i.e. how adapters are trained, how they can benefit old tasks without old data, whether the backbone is trained, and how inference-time gating works. The rebuttal directly resolved these issues, and the reviewer explicitly acknowledged that the initial concerns were addressed and increased the score likely to 8.

Reviewer 7TJs (score 4) was concerned primarily on theoretical rigor, raising concerns about unclear loss definitions, strong assumptions, and inconsistencies between the stated theorems and the paper’s claims. Despite discussion, the reviewer remained unconvinced and explicitly stated that the revised explanations increased confusion rather than resolving it. The authors made a final response by providing additional theoretical material, but considering the material, the AC thinks it is likely that the reviewer would retain the original score.

The Area Chair shares concerns about limited novelty, reliance on a memory buffer, and incremental performance improvements. While the rebuttal clearly improved clarity and presentation and convinced one reviewer, it did not resolve these main open concerns.

---

### Decision · Program_Chairs · 2026-01-26

Reject